

# A model of the weathering crust and microbial activity on an ice-sheet surface

Tilly Woods[1] and Ian J. Hewitt[1]

[1]Mathematical Institute, University of Oxford, Oxford, UK

**Correspondence:** Tilly Woods (tilly.woods@maths.ox.ac.uk)

**Abstract.** Shortwave radiation penetrating beneath an ice-sheet surface can cause internal melting and the formation of a near-surface porous layer known as the weathering crust, a dynamic hydrological system that provides home to impurities and microbial life. We develop a mathematical model, incorporating thermodynamics and population dynamics, for the evolution of such layers. The model accounts for conservation of mass and energy, for internal and surface-absorbed radiation, and for logistic growth of a microbial species mediated by nutrients that are sourced from the melting ice. It also accounts for potential melt-albedo and microbe-albedo feedbacks, through the dependence of the absorption coefficient on the porosity or microbial concentration. We investigate one-dimensional steadily melting solutions of the model, which give rise to predictions for the weathering crust depth, water content, melt rate, and microbial abundance, depending on a number of parameters. In particular, we examine how these quantities depend on the forcing energy fluxes, finding that the relative amounts of shortwave (surface-penetrating) radiation and other heat fluxes are particularly important in determining the structure of the weathering crust. The results explain why weathering crusts form and disappear under different forcing conditions, and suggest a range of possible changes in behaviour in response to climate change.

## 1 Introduction

The weathering crust on the surface of the south-western Greenland ice sheet affects the transport and storage of meltwater and microbes, as well as the albedo of the ice sheet (Muller and Keeler, 1969; Cooper et al., 2018; Irvine-Fynn et al., 2021). The weathering crust is a layer, up to about a metre thick, of very porous ice at the surface of the ice sheet, with impermeable ice below (Cooper et al., 2018). The porous structure is formed by shortwave radiation penetrating below the surface, causing "internal" melting. Most of the thickness of the weathering crust is saturated with meltwater, forming what is known as the "weathering crust aquifer" (e.g. Christner et al., 2018).

The weathering crust is home to many microorganisms, including dark-coloured glacier ice algae at the surface (following the naming convention recommended by Cook et al. (2019) and Hotaling et al. (2021)); and algae and bacteria in the weathering crust aquifer (Irvine-Fynn et al., 2021). The growth and transport of these microbes affects the cycling of nutrients such as carbon (e.g. Stibal et al., 2012) and nitrogen (e.g. Telling et al., 2012). For example, microbial activity affects whether an ice sheet surface is a source or a sink of $CO_2$; and microbes and nutrients can be transported by the flow of meltwater, impacting downstream habitats such as oceans and soils (Stibal et al., 2012).





The weathering crust can transiently store meltwater and affect how the water is transported across the ice sheet surface (Cooper et al., 2018). Moreover, the presence of the weathering crust can affect the amount of meltwater produced, since it modulates how much shortwave radiation is reflected or absorbed by the ice. For example, the weathering crust reflects more radiation than the glazed, bare ice left behind when the crust is removed (Tedstone et al., 2020). Weathering crust removal
typically occurs during periods of cloud cover, warm air temperatures and strong winds (Muller and Keeler, 1969).

In addition to changes to the ice sheet surface, biological factors - including glacier ice algae (Benning et al., 2014) - contribute to the darkening of the Greenland ice sheet (Hotaling et al., 2021), lowering the albedo and increasing melting. Work has been done to assess and quantify the contribution of glacier ice algae to albedo and surface runoff (e.g. Cook et al., 2017, 2020; Tedstone et al., 2020; Chevrollier et al., 2022). For example, Cook et al. (2020) estimated that 10-13% of the total
runoff from south-western Greenland in summer 2017 could be attributed to algae growth. However, it is difficult to isolate the biological impact from the impact due to changes to the weathering crust structure, as well as other factors including non-biological impurities (Tedesco et al., 2016; Cook et al., 2017; Tedstone et al., 2020). The potential positive feedback loop associated with glacier ice algae and melting, whereby increased melting enables more algal growth, demonstrates the intrinsic link between the structure of the weathering crust and the microbial activity within it (Tedstone et al., 2020; McCutcheon et al.,
2021). All of this emphasises the need to understand the microbial community and the ice structure as one.

In this paper, we develop a continuum model for the evolution of the weathering crust, to which we couple a model for microbes and nutrients in the weathering crust aquifer. We reduce our model to one dimension and focus on steadily melting solutions in which constant radiation and atmospheric conditions cause the surface to lower at a constant rate. We investigate how the structure of the weathering crust, the amount of melting, and the amount and distribution of microbes are affected
by the forcing conditions. In Sect. 2 we describe the model in the absence of any microbes, and we discuss solutions for the weathering crust structure. In Sect. 3 we extend the model to include microbes and nutrients, and discuss solutions for the microbial and nutrient distributions as well as the influence of radiative forcings. In Sect. 4, we use our model to investigate feedbacks between microbial growth, albedo and melting. Finally, in Sect. 5 we discuss our results and conclude.

## 2  Weathering crust model

### 2.1  Model

Firstly, we present a model for the evolution of the structure of the weathering crust, without any microbes or nutrients present. We model the weathering crust as porous ice with upper surface $z = h(t)$. The pores can contain both water and air. The ice takes up a volume fraction $1 - \phi$, where $\phi$ is the porosity. The saturation $S$ describes the proportion of the pore space taken up by water, so $\phi S$ is the volume fraction of water and $\phi(1 - S)$ is the volume fraction of air. The porosity and saturation can
evolve by the ice melting internally at a rate $m_{\mathrm{int}}$ (with negative $m_{\mathrm{int}}$ corresponding to water freezing in the pores) and by the water flowing through the porous ice. The internal melting is driven by shortwave (solar) radiation penetrating below the ice surface. We now present the governing equations.





### 2.1.1 Mass conservation

Mass conservation equations for the ice, water and air respectively are

$$\frac{\partial}{\partial t}\Big((1-\phi)\rho_{\mathrm{i}}\Big) + \nabla\cdot\Big((1-\phi)\rho_{\mathrm{i}}\mathbf{u}_{\mathrm{i}}\Big) = -m_{\mathrm{int}}, \tag{1}$$

$$\frac{\partial}{\partial t}\Big(\phi S\rho_{\mathrm{w}}\Big) + \nabla\cdot\Big(\phi S\rho_{\mathrm{w}}\mathbf{u}_{\mathrm{w}}\Big) = m_{\mathrm{int}}, \tag{2}$$

$$\frac{\partial}{\partial t}\Big(\phi(1-S)\rho_{\mathrm{a}}\Big) + \nabla\cdot\Big(\phi(1-S)\rho_{\mathrm{a}}\mathbf{u}_{\mathrm{a}}\Big) = 0, \tag{3}$$

where $\rho_{\mathrm{i}}$, $\rho_{\mathrm{w}}$, $\rho_{\mathrm{a}}$ are the densities of ice and water respectively; and $\mathbf{u}_{\mathrm{i}}$, $\mathbf{u}_{\mathrm{w}}$, $\mathbf{u}_{\mathrm{a}}$ are their velocities. Since air is much less dense than ice and water ($\rho_{\mathrm{a}} \ll \rho_{\mathrm{i}}, \rho_{\mathrm{w}}$), we can neglect the air equation (3). For our purposes in this study, we treat the ice and water velocities as prescribed (and we will later set them to zero for the specific one-dimensional solution we consider). Alternatively, a relationship between the the $\mathbf{u}_{\mathrm{i}}$ and $\mathbf{u}_{\mathrm{w}}$ could come from Darcy's law for flow through a porous medium.

### 2.1.2 Energy equation

We also need to model the temperature $T$, which we assume to be homogeneous across the ice-water-air mixture. We assume that if water is present ($\phi > 0, S > 0$) then the region is at the melting temperature $T_{\mathrm{m}}$. If there is no water present ($S = 0$), then the ice can be below the melting temperature, $T \leq T_{\mathrm{m}}$.

Conservation of energy leads to the equation

$$\frac{\partial}{\partial t}\Big[\big((1-\phi)\rho_{\mathrm{i}}c_{\mathrm{i}} + \phi S\rho_{\mathrm{w}}c_{\mathrm{w}}\big)T\Big] + \nabla\cdot\Big[\big((1-\phi)\rho_{\mathrm{i}}c_{\mathrm{i}}\mathbf{u}_{\mathrm{i}} + \phi S\rho_{\mathrm{w}}c_{\mathrm{w}}\mathbf{u}_{\mathrm{w}}\big)T\Big] = \nabla\cdot\Big[\big(k_{\mathrm{i}}(1-\phi) + k_C\phi S\big)\nabla T\Big]$$
$$-\nabla\cdot\mathbf{F} - \mathcal{L}m_{\mathrm{int}}. \tag{4}$$

The terms on the left hand side represent the rate of change and advection of heat energy. The terms on the right hand side correspond to heat conduction according to Fourier's law, the heat source provided by internal shortwave radiation $\mathbf{F}$ (discussed further below), and the energy available for melting the ice. Here, $c_{\mathrm{i}}$, $c_{\mathrm{w}}$ are the specific heat capacities and $k_{\mathrm{i}}$, $k_C$ are the thermal conductivities for ice and water respectively; and $\mathcal{L}$ is the latent heat of fusion of water. Consistent with assuming negligible density of air, we ignore both the heat content and the conductive heat flux in the air. We have assumed a simple volume averaging for the effective thermal conductivity of the combined ice-water mixture.

The energy equation captures two cases. Firstly, when the system is below the melting temperature ($T < T_{\mathrm{m}}$), there cannot be any internal melting ($m_{\mathrm{int}} = 0$) and there is no water present ($S = 0$). The energy equation (4) determines the temperature. Alternatively, when the system is at the melting temperature ($T = T_{\mathrm{m}}$), melting is possible ($m_{\mathrm{int}} \geq 0$) and water can be present ($S \geq 0$). Since we know that $T = T_{\mathrm{m}}$, the energy equation (4) instead determines the internal melt rate $m_{\mathrm{int}}$.





### 2.1.3 Radiation model


The internal melting of the ice is caused by some of the shortwave radiation from the sun penetrating and being absorbed below the surface of the ice, acting as an internal heat source. We model the internal shortwave radiation flux $\mathbf{F}$ following the two-stream approximation (e.g. Perovich, 1990; Liston et al., 1999; Taylor and Feltham, 2004, 2005). We assume that the net radiation flux can be decomposed into upward $F_+$ and downward $F_-$ travelling components. The net radiation flux is then

$\mathbf{F} = (F_+ - F_-)\hat{\mathbf{k}}$, where $\hat{\mathbf{k}}$ is the upward-pointing unit vector, with $F_\pm$ satisfying

$$\frac{\partial F_+}{\partial z} = -(\alpha + r)F_+ + rF_-, \tag{5}$$

$$\frac{\partial F_-}{\partial z} = (\alpha + r)F_- - rF_+, \tag{6}$$

where $\alpha$ and $r$ are absorption and scattering coefficients, respectively, with $\alpha, r > 0$. We assume that $\alpha$ and $r$ represent bulk
properties of the ice-water mixture and, for now, we also assume that $\alpha$ and $r$ are constant. These equations tell us that, as we move through the ice, $F_+$ is decreased by some of this radiation being absorbed and reflected, but increased by some of $F_-$ being reflected (and so changing from a downward to an upward flux). Similarly for $F_-$.

### 2.1.4 Kinematic conditions

The kinematic conditions for the ice and the water at the surface $z = h(t)$ are, respectively,

$$\rho_i(1-\phi)\left(\mathbf{u}_i \cdot \hat{\mathbf{k}} - \dot{h}\right) = \rho_w m_{\text{surf}} \quad \text{at} \quad z = h(t), \tag{7}$$

$$\rho_w \phi \left(\mathbf{u}_w \cdot \hat{\mathbf{k}} - \dot{h}\right) = \rho_w(r_{\text{off}} - m_{\text{surf}}) \quad \text{at} \quad z = h(t), \tag{8}$$

where $\mathbf{u}_i \cdot \hat{\mathbf{k}}$ and $\mathbf{u}_w \cdot \hat{\mathbf{k}}$ are the upward ice and water velocities, $\dot{h} = dh/dt$ is the velocity of the surface, $m_{\text{surf}}$ is the rate of surface melting, and $r_{\text{off}}$ is the rate of runoff.
We assume that any water at the surface drains down into the ice below, provided that the ice is unsaturated. If the surface becomes saturated, we assume that any excess water immediately runs off. (In reality, this must be assumed to happen via horizontal flow to other parts of the ice surface, and ultimately into the ocean.) The runoff term $r_{\text{off}}$ is therefore only non-zero when the surface is saturated.

### 2.1.5 Surface energy balance

The energy balance at the surface $z = h(t)$ is (see for example Hoffman et al., 2008; van den Broeke et al., 2008, 2011)

$$-\left((1-\phi)k_i + \phi S k_C\right)\frac{\partial T}{\partial z} + \chi(1-a)Q_{\text{si}} - \epsilon\sigma T^4 + Q_{\text{li}} + C(T_a - T) = \rho_w \mathcal{L} m_{\text{surf}} \quad \text{at} \quad z = h(t), \tag{9}$$





where the terms in order from left to right represent conduction of heat from the porous ice; incoming shortwave radiation absorbed at the surface ($Q_\mathrm{si}$ is the incoming shortwave (solar) radiation, $a$ is the albedo, $\chi$ is the proportion of the total shortwave radiation absorbed by the ice which is absorbed at the surface, discussed further below); emitted longwave radiation

given by the Stefan–Boltzmann law for a black body ($\epsilon$ is the surface emissivity, a correction for the fact the the ice is not a true black body, and $\sigma$ is the Stefan–Boltzmann constant); incoming longwave radiation (for example from reflection off clouds); turbulent heat transfer ($C$ is a turbulent heat transfer coefficient, and $T_\mathrm{a}$ is the reference air temperature 2m above the ice surface); and latent heat flux available for surface melting at rate $m_\mathrm{surf}$.

The turbulent heat flux term takes into account the fact that, if the air is warmer than the surface, winds can cause turbulent

eddies which transfer heat from the air to the surface. For simplicity, we are neglecting latent heat transfer due to evaporation/ablation since van den Broeke et al. (2008, 2011) showed this contribution to be small compared to the other terms, at least in the ablation zone of the west Greenland ice sheet.

Often, it is assumed that shortwave radiation incident on an ice sheet surface is either absorbed at the surface or reflected straight back out, with no radiation being transmitted below the surface. This assumption results in a $(1-a)Q_\mathrm{si}$ term in the

surface energy balance. However, a key part of weathering crust formation is the penetration of shortwave radiation below the ice surface. Hence, based on methods used by Hoffman et al. (2014) and Law et al. (2020), we introduce the parameter $\chi < 1$ - the proportion of the total amount of shortwave radiation absorbed by the ice which is absorbed at the surface - to allow some of the shortwave radiation to penetrate into the subsurface ice, where we model it explicitly as the internal shortwave radiation flux $\mathbf{F}$ (see discussion around equations (5) and (6)). In particular, this means that the downward internal radiation

flux $F_-$, at the surface $z = h(t)$, is $\left(1 - \chi(1-a)\right)Q_\mathrm{si}$. The upward flux $F_+$ is $aQ_\mathrm{si}$. Note that in this model the albedo $a$ is not an independent parameter. It is a result of the internal radiation calculation, dependent on the scattering and absorption coefficients $\alpha$ and $r$ (see Sect. 2.3).

By linearising about the melting temperature, the surface energy balance can be re-written as

$$-\Big((1-\phi)k_\mathrm{i} + \phi k_C\Big)\frac{\partial T}{\partial z} + \chi(1-a)Q_\mathrm{si} + Q_0 - \upsilon(T - T_\mathrm{m}) = \rho_\mathrm{w}\mathcal{L}m_\mathrm{surf} \quad \text{at} \quad z = h(t), \tag{10}$$

where $Q_0 = Q_\mathrm{li} - \epsilon\sigma T_\mathrm{m}^4 + C(T_\mathrm{a} - T_\mathrm{m})$ is a combination of longwave radiation and turbulent heat flux, which we will prescribe as a function of time; and $\upsilon = C + 4\epsilon\sigma T_\mathrm{m}^3$ is an effective heat transfer coefficient, combining turbulence and radiation.

### 2.1.6 Model summary

The full weathering crust model is given by the mass conservation equations (1) and (2) for ice and water, the energy equation (4), and the two-stream equations (5) and (6) for the internal shortwave radiation, along with a prescription of the ice and

water velocities $\mathbf{u}_\mathrm{i}$ and $\mathbf{u}_\mathrm{w}$, and suitable boundary conditions. As well as the kinematic conditions (7) and (8), the surface energy balance (10), and the condition for downward-travelling radiation $F_-$ at the surface, we also require a condition on the upward-travelling radiation $F_+$ and the temperature $T$ in the far-field (discussed below). The incoming shortwave radiation $Q_\mathrm{si}$ and the "other" surface radiation $Q_0$ (a combination of longwave radiation and turbulent heat fluxes) are prescribed functions of time, representing the climate forcing. As well as the temperature $T$, porosity $\phi$ and saturation $S$, additional outputs from





| Quantity | Symbol | Default value | Units |
|---|---|---|---|
| Density of ice | $\rho$ | 910 | kg m$^{-3}$ |
| Specific heat capacity of ice at 0°C | $c$ | 2097 | J kg$^{-1}$ K$^{-1}$ |
| Thermal conductivity | $k$ | 2.1 | kg m$^{-3}$ K$^{-1}$ |
| Melting temperature | $T_{\mathrm{m}}$ | 273.15 | K |
| Latent heat of melting | $\mathcal{L}$ | 334000 | m$^2$ s$^{-2}$ |
| Albedo | $a$ | 0.6 | |
| Proportion of radiation absorbed at the surface[a] | $\chi$ | 0.36 | |
| Extinction coefficient of internal radiation[b] | $\kappa$ | 1.5 | m$^{-1}$ |
| Unnamed optical property[c] | $s$ | 0.7009 | |
| Absorption coefficient[c] | $\alpha$ | 0.2637 | m$^{-1}$ |
| Scattering coefficient[c] | $r$ | 4.1338 | m$^{-1}$ |
| Proportion of solar radiation that is PAR[d] | $\alpha_{\mathrm{PAR}}$ | 0.56 | |
| Maximum microbial growth rate | $\beta_A$ | 20 | d$^{-1}$ |
| Maximum nutrient uptake rate | $\beta_C$ | $10^{-6}$ | $\mu$mol cell$^{-1}$ d$^{-1}$ |
| Measure of nutrient limitation | $k_C$ | 1 | $\mu$mol L$^{-1}$ |
| Measure of radiation limitation | $k_F$ | 100 | W m$^{-2}$ |
| Maximum microbial abundance[e] | $A_{\mathrm{max}}$ | $10^4$ | cells mL$^{-1}$ |
| Death rate of microbes in water | $d_{\mathrm{w}}$ | 0 | d$^{-1}$ |
| Death rate of microbes in ice | $d_{\mathrm{i}}$ | 0 | d$^{-1}$ |
| Incoming shortwave radiation flux | $Q_{\mathrm{si}}$ | 200 | W m$^{-2}$ |
| Turbulent heat and longwave radiation flux[f] | $Q_0$ | -20 | W m$^{-2}$ |
| Temperature at depth | $T_\infty$ | -10 | °C |
| Microbial abundance at in deep ice[e] | $A_\infty$ | $10^2$ | cells mL$^{-1}$ |
| Nutrient concentration in deep ice[g] | $C_\infty$ | 1 | $\mu$mol L$^{-1}$ |

**Table 1.** Default values of the parameters in the model. Unless otherwise stated, these are the parameter values used. [a]Motivated by Hoffman et al. (2014). [b]Taylor and Feltham (2005). [c]Calculated from $\kappa$, $a$ and $\chi$ using (24) and definitions of $\kappa$ and $s$. [d]Jørgensen and Bendoricchio (2001). [e]Motivated by Christner et al. (2018). [f]Inferred from van den Broeke et al. (2011). [g]Motivated by Holland et al. (2019). Many of the ice properties are taken from Cuffey and Paterson (2010).

the model are the surface and internal melt rates $m_{\mathrm{surf}}$ and $m_{\mathrm{int}}$, and surface runoff $r_{\mathrm{off}}$. The default parameter values used in the model are shown in Table 1.

## 2.2  Steadily melting states

We now make some simplifications to our model. Firstly, we make the approximation that the ice and water have the same properties, introducing a shared density $\rho = \rho_{\mathrm{i}} = \rho_{\mathrm{w}}$, thermal conductivity $k = k_{\mathrm{i}} = k_C$ and specific heat capacity $c = c_{\mathrm{i}} = c_{\mathrm{w}}$.



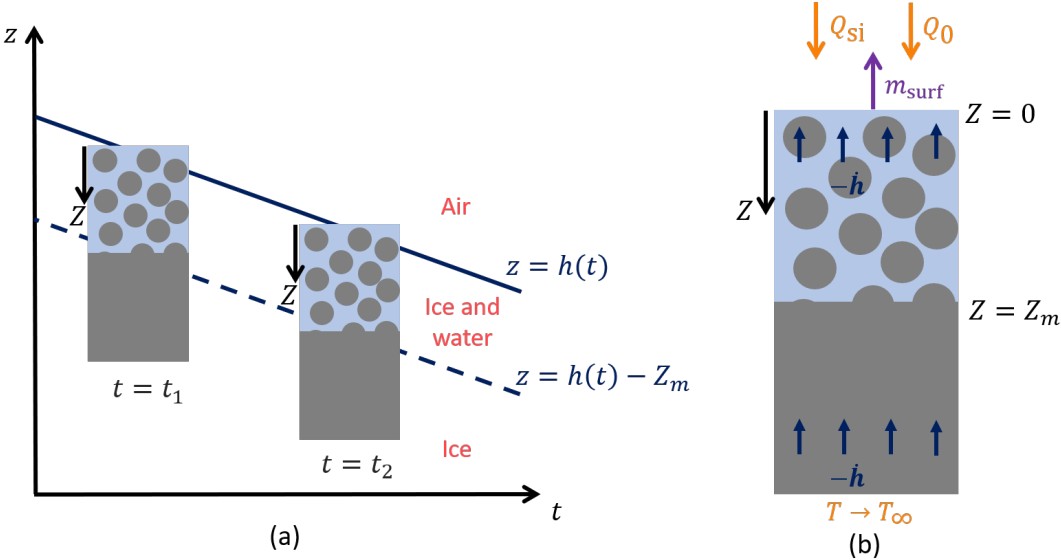

**Figure 1.** Schematic of the steadily melting setup. (a) shows how the surface of the ice $z = h(t)$ lowers over time in the steadily melting state, with the ice melting in such a way that it looks like the ice column is simply being translated downwards. (b) shows this column of ice in the $Z = z - h(t)$ frame. The forcings $Q_{si}, Q_0, T_\infty$ and the surface melt rate $m_{surf}$ are also shown.

Further, we reduce our model to one dimension, considering a vertical column of ice. We also assume that the porous ice is saturated ($S = 1$) and that there is no flow of ice or water ($\mathbf{u}_i = \mathbf{u}_w = \mathbf{0}$). Some of these simplifications could quite easily be relaxed, but making them allows for a more transparent interpretation of the model solutions.

We now seek what we will refer to as "steadily melting" solutions. These are solutions in which the surface is lowering at a constant rate $(-\dot{h})$, with the subsurface ice melting at exactly a rate $m_{int}$ to maintain a steady porosity profile below the 155 surface. Therefore, relative to the surface, the solution looks the same at each snapshot in time, as if our column of porous ice is simply being translated downwards, as shown in Fig. 1 (a). However, this effect is produced by internal and surface melting, not by the ice moving (recall that the ice velocity is zero, by assumption; in reality the downward melting of the ice surface may be counteracted by the upward advection of ice from the interior of the ice sheet).

The steadily melting setup suggests a natural change of coordinates to

$$Z = h(t) - z, \tag{11}$$

the depth below the surface $z = h(t)$. Note that $Z$ points downwards. In the $Z$ frame, the melting surface of the ice is fixed at $Z = 0$, and the ice below gets advected upwards towards the surface at rate $-\dot{h}$, as shown in Fig. 1 (b). Making this transformation, and using the simplifications previously discussed, we can turn our general weathering crust equations into equations for the steadily melting solution. The two-stream equations (5) and (6) for the internal radiation $F = F_- - F_+$ become

$$\frac{dF_+}{dZ} = (\alpha + r)F_+ - rF_-, \tag{12}$$





$$\frac{\mathrm{d}F_-}{\mathrm{d}Z} = -(\alpha + r)F_- + rF_+. \tag{13}$$

Both the ice and water mass conservation equations (1) and (2) reduce to

$$\rho \dot{h} \frac{\mathrm{d}\phi}{\mathrm{d}Z} = m_{\mathrm{int}}, \tag{14}$$

and the energy equation (4) becomes

$$\rho c \dot{h} \frac{\mathrm{d}T}{\mathrm{d}Z} = k \frac{\mathrm{d}^2 T}{\mathrm{d}Z^2} - \frac{\mathrm{d}F}{\mathrm{d}Z} - \mathcal{L} m_{\mathrm{int}}. \tag{15}$$

The advection terms $\dot{h}\mathrm{d}/\mathrm{d}Z$ come from the time derivatives under the transformation $z \to Z$. As for boundary conditions, the kinematic conditions (7) and (8) tell us that

$$-\dot{h}(1-\phi) = m_{\mathrm{surf}} \quad \mathrm{at} \quad Z = 0, \tag{16}$$

and that the runoff rate is

$$r_{\mathrm{off}} = -\dot{h}. \tag{17}$$

The linearised surface energy balance (10) becomes

$$k\frac{\mathrm{d}T}{\mathrm{d}Z} + \chi(1-a)Q_{\mathrm{si}} + Q_0 = \rho \mathcal{L} m_{\mathrm{surf}} \quad \mathrm{at} \quad Z = 0, \tag{18}$$

where we have used the fact that $T = T_{\mathrm{m}}$ at the surface in order for surface melting to be possible. In addition to these, we also

prescribe a cold temperature $T_\infty < T_{\mathrm{m}}$ at depth, so

$$T \to T_\infty \quad \mathrm{as} \quad Z \to \infty. \tag{19}$$

Finally, the boundary conditions for the two-stream equations (12) and (13) become

$$F_- = (1 - \chi(1-a))Q_{\mathrm{si}} \quad \mathrm{at} \quad Z = 0, \tag{20}$$

and

$\quad F_+ \to 0 \quad \mathrm{as} \quad Z \to \infty, \tag{21}$

since we assume there is no source of shortwave radiation at depth. We also have

$$F_+ = aQ_{\mathrm{si}} \quad \mathrm{at} \quad Z = 0, \tag{22}$$

which serves to determine the albedo $a$ (which is defined as the overall ratio of the reflected and incident shortwave radiation). Note that in some cases it may be more natural to prescribe the albedo $a$ as a known (ie. measured) quantity. In this model,





the albedo is related to $\chi$, $\alpha$ and $r$, so independently prescribing $a$ would mean that one of these other parameters must be prescribed consistently (see (24) below).

There are three forcings in our model, which we prescribe. These are the incoming shortwave radiation $Q_{\mathrm{si}}$, the combined longwave radiation and turbulent heat fluxes $Q_0$, and the temperature of the ice at depth, $T_\infty$. The outputs of the model are the internal shortwave radiation $F$, porosity $\phi$ and temperature $T$, as well as the internal melt rate $m_{\mathrm{int}}$, surface melt rate $m_{\mathrm{surf}}$, runoff $r_{\mathrm{off}}$ and rate of change of surface height $-\dot{h}$. We choose values of the forcings such that the surface is melting ($m_{\mathrm{surf}} > 0$) and the deep ice is cold. Therefore, as shown in Fig. 1, we expect there to be a temperate region $0 < Z < Z_{\mathrm{m}}$ near the surface where the ice is porous ($\phi > 0$) and at the melting temperature ($T = T_{\mathrm{m}}$). In $Z > Z_{\mathrm{m}}$, the ice is solid ($\phi = 0$) and cold ($T < T_{\mathrm{m}}$). We refer to $Z_{\mathrm{m}}$ as the melting depth, which must also be determined by the model. Note that having impermeable, solid ice at $Z = Z_{\mathrm{m}}$ combined with our assumption that the ice is saturated with water justifies our assumption of no water flow ($\mathbf{u}_{\mathrm{w}} = \mathbf{0}$), at least in the vertical direction. Furthermore, Irvine-Fynn et al. (2021) have observed lateral flow through the weathering crust to be slow (less than about 0.13 m d$^{-1}$), which somewhat justifies neglecting horizontal water velocities and using a one-dimensional model.

## 2.3 Results

In the case where we have constant absorption coefficient $\alpha$, scattering coefficient $r$ and albedo $a$, we can find the steadily melting solution analytically. Firstly, the two-stream equations (12) and (13) can be solved with boundary conditions (20) and (21) to find that the the internal radiation $F = F_- - F_+$ decreases exponentially with depth,

$$F_- = \big(1 - \chi(1-a)\big)Q_{\mathrm{si}}e^{-\kappa Z}, \quad F_+ = sF_-, \tag{23}$$

where the extinction coefficient is defined as $\kappa = \sqrt{\alpha^2 + 2\alpha r}$, $s = (\kappa - \alpha)/(\kappa + \alpha)$, and the extra boundary condition (22) for $F_+$ tells us that

$$a = s\big(1 - \chi(1-a)\big). \tag{24}$$

This is the relationship telling us that we can only prescribe three of $\alpha$, $r$, $a$ and $\chi$ (or equivalently, three of $\kappa$, $s$, $a$ and $\chi$). Using the condition (24), we can eliminate $s$ from $F_+$ to obtain

$$F = F_- - F_+ = (1-a)(1-\chi)Q_{\mathrm{si}}e^{-\kappa Z}. \tag{25}$$

This is the familiar Beer's law, previously used to model radiation attenuation in sea ice (e.g. Maykut and Untersteiner, 1971) and glaciers (e.g. Kelliher et al., 1996). We chose to use the two-stream approximation to model the internal radiation rather than assuming Beer's law straight away so that our model can later be extended to non-constant absorption and scattering coefficients $\alpha$ and $r$.

We can now analytically solve for the steadily melting temperature $T$, porosity $\phi$ and internal melt rate $m_{\mathrm{int}}$. As mentioned previously, we seek solutions with a temperate region $0 < Z < Z_{\mathrm{m}}$ above a cold region $Z > Z_{\mathrm{m}}$. The boundary conditions



between the two regions are

$$\phi = 0, \quad T = T_{\mathrm{m}}, \quad \frac{\mathrm{d}T}{\mathrm{d}Z} = 0 \quad \text{at} \quad Z = Z_{\mathrm{m}}, \tag{26}$$

where the final condition comes from continuity of conductive heat flux. Also, since $T = T_{\mathrm{m}}$ everywhere in $0 < Z < Z_{\mathrm{m}}$, there is no heat flux at the surface, so the surface energy balance (18) directly tells us the surface melt rate,

$$m_{\mathrm{surf}} = \frac{\chi(1-a)Q_{\mathrm{si}} + Q_0}{\rho\mathcal{L}}. \tag{27}$$

In the cold region, we integrate the temperature equation (4) and apply the far field condition (19), along with the fact that a uniform far-field temperature means that the heat flux tends to zero, to get the global energy balance

$$\rho c \dot{h} T - k\frac{\mathrm{d}T}{\mathrm{d}Z} + F = \rho c \dot{h} T_\infty, \tag{28}$$

for $Z > Z_{\mathrm{m}}$. Evaluating this at $Z = Z_{\mathrm{m}}$ and using the boundary conditions (26) gives us an expression which can be rearranged to tell us $Z_{\mathrm{m}}$ in terms of $\dot{h}$,

$$Z_{\mathrm{m}} = -\frac{1}{\kappa}\log\left(\frac{-\rho c \dot{h}(T_{\mathrm{m}} - T_\infty)}{(1-\chi)(1-a)Q_{\mathrm{si}}}\right). \tag{29}$$

The global energy balance (28) can be integrated once more, using the boundary condition (26) for $T$, to find that the temperature is

$$T = \begin{cases} T_{\mathrm{m}}, & 0 < Z < Z_{\mathrm{m}}, \\ T_{\mathrm{m}} - \frac{(1-\chi)(1-a)Q_{\mathrm{si}}}{k\kappa + \rho c \dot{h}}\left(e^{-\kappa Z} - e^{-\kappa Z_{\mathrm{m}}} e^{\frac{\rho c \dot{h}}{k}\left(Z - Z_{\mathrm{m}}\right)}\right) + T_\infty\left(1 - e^{\frac{\rho c \dot{h}}{k}\left(Z - Z_{\mathrm{m}}\right)}\right), & Z > Z_{\mathrm{m}}. \end{cases} \tag{30}$$

In the temperate region, the energy equation (15) determines the internal melt rate $m_{\mathrm{int}}$ from the internal radiation $F$ as

$$m_{\mathrm{int}} = \begin{cases} \frac{\kappa(1-\chi)(1-a)Q_{\mathrm{si}}}{c\mathcal{L}} e^{-\kappa Z}, & 0 < Z < Z_{\mathrm{m}}, \\ 0, & Z > Z_{\mathrm{m}}. \end{cases} \tag{31}$$

In turn, $\phi$ can be calculated from $m_{\mathrm{int}}$ using the mass conservation equation (14), giving

$$\phi = \begin{cases} -\frac{(1-\chi)(1-a)Q_{\mathrm{si}}}{\rho c \dot{h}\mathcal{L}}\left(e^{-\kappa Z} - e^{-\kappa Z_{\mathrm{m}}}\right), & 0 < Z < Z_{\mathrm{m}}, \\ 0, & Z > Z_{\mathrm{m}}. \end{cases} \tag{32}$$

Finally, the kinematic condition (16) determines the rate of change of surface height $\dot{h}$ as

$$\dot{h} = -\frac{(1-a)Q_{\mathrm{si}} + Q_0}{\rho\mathcal{L} + \rho c(T_{\mathrm{m}} - T_\infty)}, \tag{33}$$

which can be substituted into the expression (29) for the melting depth to give

$$Z_{\mathrm{m}} = \frac{1}{\kappa}\log\left(\left(\frac{(1-a)(1-\chi)Q_{\mathrm{si}}}{\rho c(T_{\mathrm{m}} - T_\infty)}\right)\middle/\left(\frac{(1-a)Q_{\mathrm{si}} + Q_0}{\rho\mathcal{L} + \rho c(T_{\mathrm{m}} - T_\infty)}\right)\right). \tag{34}$$



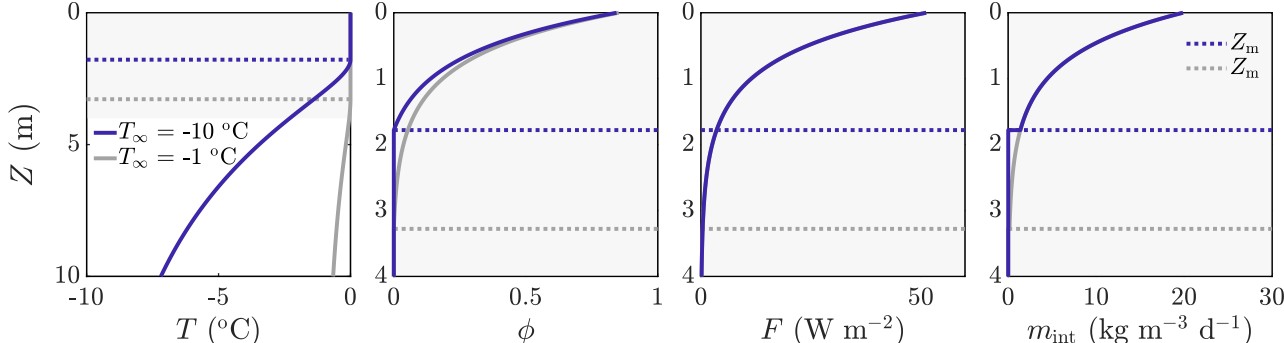

**Figure 2.** Example solution for $T_\infty = -10°C$ (dark blue) and $T_\infty = -1°C$ (grey). The dotted lines show the melting depth $Z_m$, with $Z_m \approx 1.8$ m when $T_\infty = -10°C$ and $Z_m \approx 3.3$ m when $T_\infty = -1°C$. The rate of change of surface height is $\dot{h} \approx -1.6$ cm d$^{-1}$ when $T_\infty = -10°C$ and $\dot{h} \approx -1.7$ cm d$^{-1}$ when $T_\infty = -1°C$. The surface melt rate is $m_{surf} \approx 0.25$ cm d$^{-1}$ in both cases. The other parameter values are those shown in Table 1. The shaded region on the left panel corresponds to the smaller region shown in the other panels.

As automatic from (17), the surface runoff in this situation is equal to the rate of surface lowering, $r_{off} = -\dot{h}$. Example solutions for temperature, porosity, internal radiation and internal melt rate are shown in Fig. 2 for two different values of the deep-ice temperature $T_\infty$. Unsurprisingly, a colder deep-ice temperature (blue lines) leads to a smaller porous region, with the dotted

lines indicating $Z = Z_m$.

For our assumption that the surface is melting to be valid, we need the surface melt rate $m_{surf}$ to be positive. Using the expression (27) for $m_{surf}$, this gives a restriction on the possible values of the prescribed radiations $Q_{si}$ and $Q_0$,

$$\chi(1-a)Q_{si} + Q_0 > 0. \tag{35}$$

Furthermore, the solution found here, with a porous ice region occupying $0 < Z < Z_m$, is only valid if the calculated $Z_m$ is

positive, and hence the argument of the logarithm in (34) must be greater than 1. Physically, this can be interpreted as requiring that the raising of the internal temperature dominates the surface lowering, as we will now explain.

Firstly, as more of the internal ice is raised to the melting temperature, the size of the porous region (and hence $Z_m$) increases. This is represented by the term

$$\frac{(1-a)(1-\chi)Q_{si}}{\rho c(T_m - T_\infty)} \tag{36}$$

in the expression (34) for $Z_m$. The numerator is the amount of radiation absorbed internally - that is the energy available to heat and melt the internal ice. The denominator represents the energy needed to raise the ice temperature from $T_\infty$ at depth to the melting temperature - that is the energy required to raise the internal ice to the melting temperature. Therefore, the fraction (36) can be thought of as a measure of raising the internal ice to the melting temperature.





On the other hand, as the surface lowers, the size of the porous region (and hence $Z_\mathrm{m}$) decreases, since the porous ice at the

surface gets removed as meltwater. Surface lowering is represented by the term

$$\frac{(1-a)Q_\mathrm{si} + Q_0}{\rho\mathcal{L} + \rho c(T_\mathrm{m} - T_\infty)} \tag{37}$$

in the expression (34) for $Z_\mathrm{m}$. The numerator is the total amount of radiation absorbed by the ice, accounting for both internal

and surface absorption. This is the total energy available for melting the ice, both at the surface and internally. The denominator

represents the energy required to melt cold ice; the temperature must first be raised from $T_\infty$ to $T_\mathrm{m}$, then latent heat is needed

to effect the phase change. Indeed, it is this fraction that also gives the surface lowering rate $-\dot{h}$ in (33). (It is different from

the surface *melt* rate $m_\mathrm{surf}$, since melting occurs in two stages: first the ice can melt internally, lowering the porosity; then the

remaining ice melts at the surface.)

As previously mentioned, we expect $Z_\mathrm{m} > 0$ when raising the internal temperature dominates surface lowering. Using the

above interpretations of the fractions (36) and (37), this happens when

$$\frac{(1-a)(1-\chi)Q_\mathrm{si}}{\rho c(T_\mathrm{m} - T_\infty)} > \frac{(1-a)Q_\mathrm{si} + Q_0}{\rho\mathcal{L} + \rho c(T_\mathrm{m} - T_\infty)}, \tag{38}$$

and (34) then gives $Z_\mathrm{m} > 0$.

The conditions (35) and (38) tell us the range of parameter values for which we can have steadily melting solutions with a

non-trivial porous region, ie. for which a weathering crust exists. Figure 3(a) illustrates this graphically, showing the range of

values of shortwave radiation $Q_\mathrm{si}$ and combined longwave radiation and turbulent heat fluxes $Q_0$ for which the steadily melting

states with a non-trivial weathering crust are possible (blue to yellow region). For a fixed amount of shortwave radiation $Q_\mathrm{si}$,

increasing the other surface radiation $Q_0$ increases the amount of surface lowering but not the internal heating. This is because

$Q_0$ is completely absorbed at the surface. If $Q_0$ is too large, surface lowering is so large that ice gets advected to the surface so

quickly that it does not have the chance to get heated up to the melting temperature internally first. Hence there is no internal

melting. If $Q_0$ is too small, there is not enough energy to melt the surface.

Figure 3(b) and (c) show how the melting depth $Z_\mathrm{m}$ depends on the incoming shortwave radiation $Q_\mathrm{si}$. There is a qualitative

difference between the behaviour of $Z_\mathrm{m}$ with $Q_\mathrm{si}$ depending on the sign of $Q_0$. We see that $Z_\mathrm{m}$ increases with $Q_\mathrm{si}$ when $Q_0 > 0$

but decreases with $Q_\mathrm{si}$ when $Q_0 < 0$. The reason for this relates back to our interpretation of $Z_\mathrm{m}$ being determined by a balance

between internal heating and surface lowering. If $Q_0 < 0$, doubling $Q_\mathrm{si}$ doubles the internal heating term (36) and more than

doubles the surface lowering term (37). Therefore, as $Q_\mathrm{si}$ increases, surface lowering becomes more significant compared to

internal heating, so the porous region (and hence $Z_\mathrm{m}$) decreases. On the other hand, if $Q_0 > 0$, surface lowering becomes less

significant as $Q_\mathrm{si}$ increases, so the porous region (and hence $Z_\mathrm{m}$) increases. In both cases, a weathering crust can only exist

if the shortwave radiation $Q_\mathrm{si}$ is large enough. Physically, positive $Q_0$ corresponds to conditions of warm air temperature,

strong wind, and cloud cover, when the ice surface would melt even in the absence of solar radiation. Such conditions favour

removal of the weathering crust (Muller and Keeler, 1969), as we observe in Fig. 3 when $Q_0$ becomes too large. Negative $Q_0$

corresponds to conditions when the surface would freeze in the absence of solar radiation. In the ablation zone of the west

Greenland ice sheet, we would typically expect the net longwave radiation to be negative and the turbulent heat fluxes to be




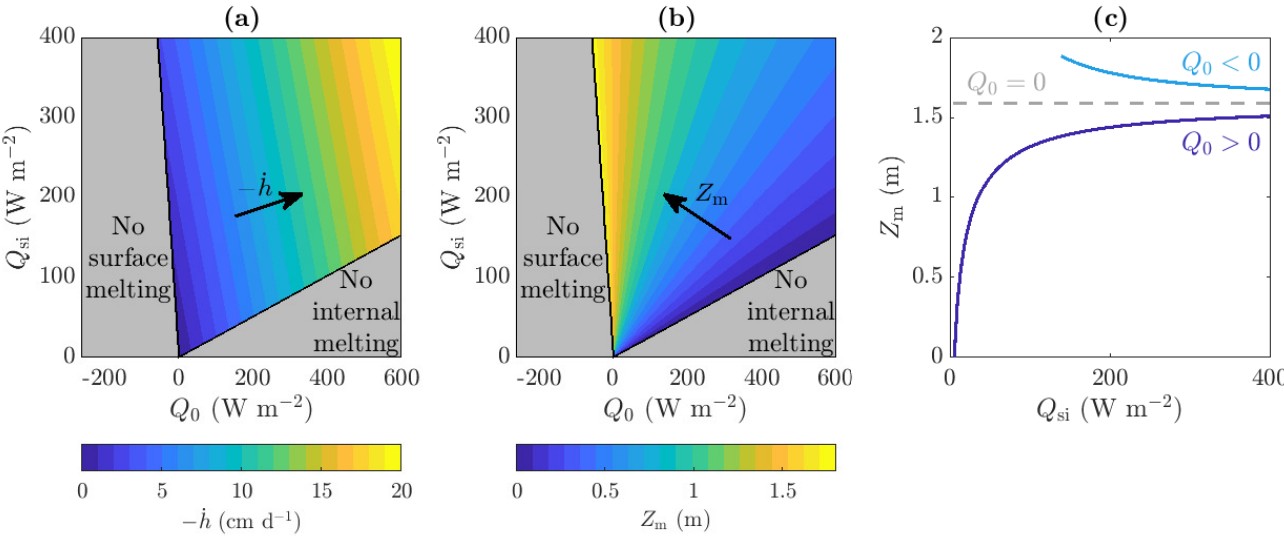

**Figure 3.** (a), (b): The range of $Q_{\text{si}}$ and $Q_0$ values for which steadily melting states with both internal and surface melting are possible, as well as the regions for which there is no surface melting and no internal melting (grey). Contours of (a) $\dot{h}$ and (b) $Z_{\text{m}}$ are shown, ranging from $\dot{h} = -20$ cm d$^{-1}$ to $\dot{h} = 0$ cm d$^{-1}$ and from $Z_{\text{m}} = 0$ m to $Z_{\text{m}} = 1.9$ m with 1 cm d$^{-1}$ and 0.1 m spacing between contours, respectively. (c): The dependence of the melting depth $Z_{\text{m}}$ on the shortwave radiation $Q_{\text{si}}$ when $Q_0 = 20$ W m$^{-2}$, $Q_0 = -20$ W m$^{-2}$ and $Q_0 = 0$ W m$^{-2}$. The other parameter values are those shown in Table 1.

positive, with the combination giving $Q_0$ as slightly negative (van den Broeke et al., 2008, 2011). An exception is in regions near the margin where ice-free tundra causes positive turbulent heat fluxes to dominate (van den Broeke et al., 2008, 2011). However, the weather conditions on the Greenland ice sheet are very changeable (e.g. Fausto et al., 2021), rapidly changing between clear, sunny conditions that favour weathering crust formation and cloudy, windy, warm weather that favours removal of the crust (Muller and Keeler, 1969). Therefore, we choose to investigate both positive and negative $Q_0$ to explore the different behaviours exhibited during these changeable conditions. The sensitivity of the weathering crust thickness $Z_{\text{m}}$ to the values of $Q_{\text{si}}$ and $Q_0$, demonstrated by our model, might partly explain why the weathering crust tends to form and be removed by changing weather conditions.

## 3 Microbes and nutrients model

### 3.1 Model

We now extend our model to include microbes and nutrients. In order to keep things simple, we model a single "representative" nutrient, the concentration of which influences the microbial growth. In reality, there may be multiple such nutrients, such as carbon, nitrogen and phosphorous (Hessen et al., 2013). The microbial growth will also depend on the shortwave radiation flux





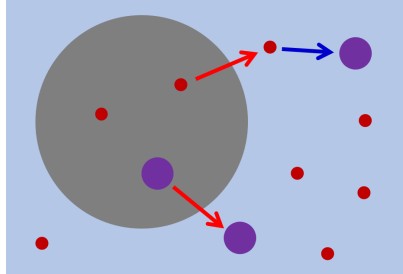

**Figure 4.** Schematic illustration of how microbes (big purple dots) and nutrients (small red dots) are included in the model. Both are transferred from ice (grey) to water (pale blue) as the ice melts (red arrows), and the microbes in the water then consume nutrients in the water (dark blue arrow).

$F$. We also model a single representative type of phototrophic microbe (a microbe that obtains energy from solar radiation), rather than considering separately different primary and secondary producers (see later discussion). We assume that both the microbes and the nutrient can exist in both the ice and the water, and when the ice melts, they are transferred from the ice to the water. Microbes in the water can consume nutrients from the water, but not from the ice, and microbes in the ice cannot consume any nutrients (see Fig. 4). Therefore, only microbes in the water are able to proliferate. We model the microbial

abundance of microbes in the water $A_\mathrm{w}$ and ice $A_\mathrm{i}$ (cells mL$^{-1}$) separately, and similarly for the nutrient concentrations $C_\mathrm{w}$ and $C_\mathrm{i}$ ($\mu$mol L$^{-1}$).

     To model the water microbial abundance $A_\mathrm{w}$, we adapt a simple logistic growth model (Murray, 1993),

$$\frac{\partial}{\partial t}\Big(\phi A_\mathrm{w}\Big) + \nabla \cdot \Big(\phi A_\mathrm{w}\mathbf{u}_\mathrm{w}\Big) = g(F,\phi C_\mathrm{w})\phi A_\mathrm{w}\left(1 - \frac{A_\mathrm{w}}{A_\mathrm{max}}\right) - d_\mathrm{w}\phi A_\mathrm{w} + \frac{A_\mathrm{i}}{\rho_\mathrm{i}}\max(m_\mathrm{int},0) - \frac{A_\mathrm{w}}{\rho_\mathrm{w}}\min(m_\mathrm{int},0), \tag{39}$$

On the left-hand side, we have the time derivative and an advection term, with the microbes being advected at the water

velocity $\mathbf{u}_\mathrm{w}$. On the right-hand side we have a growth term, a death term (with death rate $d_\mathrm{w}$), and source/sink terms due to melting/freezing. The growth term, of the form $g\phi A_\mathrm{w}(1 - A_\mathrm{w}/A_\mathrm{max})$, is the standard form for logistic growth, with $A_\mathrm{max}$ being the maximum microbial abundance. We base the form of the logistic growth rate $g$ on that used by Jørgensen and Bendoricchio (2001) for algal growth, in which $g$ depends on the intensity of photosynthetically active radiation, the volume-averaged nutrient concentration $\phi C_\mathrm{w}$ and the temperature, since these are limiting factors for growth of photosynthetic microbes. (We can ignore

the temperature dependence, since we assume that the water is always at the melting temperature.) We follow Jørgensen and Bendoricchio (2001) in using a Michaelis-Menten form of the growth term,

$$g(F,\phi C_\mathrm{w}) = \beta_A\left(\frac{\alpha_\mathrm{PAR}F}{k_\mathrm{PAR} + \alpha_\mathrm{PAR}F}\right)\left(\frac{\phi C_\mathrm{w}}{k_C + \phi C_\mathrm{w}}\right), \tag{40}$$

where $\alpha_\mathrm{PAR} = 0.56$ is the proportion of the total incident solar radiation that is photosynthetically active (ie. that can be used for photosynthesis), and $k_\mathrm{PAR}$ and $k_C$ are constants used to represent the fact that, above a certain limit, increasing the

radiation or nutrient concentration further will no longer lead to increased algal growth. $\beta_A$ is also a constant, which represents the growth rate if there is abundant radiation and nutrients (in which case the two bracketed factors are essentially 1). We refer to $\beta_A$ as the growth rate constant.





The ice microbial abundance equation is simpler. We assume that the microbes in the ice cannot grow, but can die (with death rate $d_i$), and are fixed within the ice (so they are advected at the ice velocity $\mathbf{u}_i$). Hence $A_i$ satisfies

$$\frac{\partial}{\partial t}\Big((1-\phi)A_i\Big) + \nabla \cdot \Big((1-\phi)A_i\mathbf{u}_i\Big) = -d_i(1-\phi)A_i - \frac{A_i}{\rho_i}\max(m_{\mathrm{int}},0) + \frac{A_w}{\rho_w}\min(m_{\mathrm{int}},0). \tag{41}$$

We expect that $d_i \geq d_w$, meaning that the death rate of microbes in the ice is greater than the death rate in the water.

Nutrients are transported through the weathering crust aquifer, often from nutrient-rich cryoconite holes (Cook et al., 2016). To model the nutrient transport in the water, we use similar ideas to those used to model bioreactors (e.g. Shakeel et al., 2013). The nutrients in the water can be advected and consumed by microbes. The melting of ice provides a source of nutrients into the water (and a sink from the ice), and water freezing provides a corresponding sink. The nutrients in the ice are advected with the ice, but cannot be consumed because microbes cannot access the nutrients in the ice. The equations describing this are

$$\frac{\partial}{\partial t}\Big(\phi C_w\Big) + \nabla \cdot \Big(\phi C_w\mathbf{u}_w\Big) = -\beta_C\phi A_w\left(\frac{\alpha_{\mathrm{PAR}}F}{k_{\mathrm{PAR}}+\alpha_{\mathrm{PAR}}F}\right)\left(\frac{\phi C_w}{k_C+\phi C_w}\right) + \frac{C_i}{\rho_i}\max(m_{\mathrm{int}},0) - \frac{C_w}{\rho_w}\min(m_{\mathrm{int}},0), \tag{42}$$

for the nutrients in the water, and

$$\frac{\partial}{\partial t}\Big((1-\phi)C_i\Big) + \nabla \cdot \Big((1-\phi)C_i\mathbf{u}_i\Big) = -\frac{C_i}{\rho_i}\max(m_{\mathrm{int}},0) + \frac{C_w}{\rho_w}\min(m_{\mathrm{int}},0), \tag{43}$$

for the nutrients in the ice. The uptake term has a similar form to the growth rate in (39), since we expect the consumption of nutrients to be related to the growth rate of the microbes. The constant $\beta_C$ represents a maximum nutrient uptake rate, and controls the degree to which microbes influence the nutrient concentration.

Equations (39), (41), (42) and (43) for the microbial abundances $A_w$ and $A_i$ and nutrient concentrations $C_w$ and $C_i$ are coupled to equations (1), (2), (4), (5) and (6) for the weathering crust evolution, since they depend on both porosity $\phi$ and radiation $F$.

## 3.2 Steadily melting equations

We now return to the steadily melting states considered in Sect. 2.2. Under these same assumptions, and making the same change of coordinates (11), equations (39) and (42) for the microbial abundance and nutrient concentration in the water become

$$\dot{h}\frac{\mathrm{d}}{\mathrm{d}Z}\Big(\phi A_w\Big) = \beta_A\phi A_w\left(\frac{\alpha_{\mathrm{PAR}}F}{k_{\mathrm{PAR}}+\alpha_{\mathrm{PAR}}F}\right)\left(\frac{\phi C_w}{k_C+\phi C_w}\right)\left(1-\frac{A_w}{A_{\max}}\right) - d_w\phi A_w + \frac{A_i m_{\mathrm{int}}}{\rho}, \tag{44}$$

$$\dot{h}\frac{\mathrm{d}}{\mathrm{d}Z}\Big(\phi C_w\Big) = -\beta_C\phi A_w\left(\frac{\alpha_{\mathrm{PAR}}F}{k_{\mathrm{PAR}}+\alpha_{\mathrm{PAR}}F}\right)\left(\frac{\phi C_w}{k_C+\phi C_w}\right) + \frac{C_i m_{\mathrm{int}}}{\rho}. \tag{45}$$

As an additional simplification, we make the assumption that the microbes in the water and in the ice do not die, so $d_w = d_i = 0$. In this particular situation, equations (41) and (43) for $A_i$ and $C_i$ simply tell us that the microbial abundance and nutrient concentration in the ice are constant. Since we prescribe the microbial abundance and nutrient concentration in the deep ice to be $A_\infty$ and $C_\infty$, respectively, this leads to $A_i = A_\infty$ and $C_i = C_\infty$ everywhere.





Referring back to Fig. 1, water is only present in the porous region $0 < Z < Z_\mathrm{m}$, so it is in this region that we solve equations (44) and (45) for $A_\mathrm{w}$ and $C_\mathrm{w}$. For continuity at the interface $Z = Z_\mathrm{m}$ between the porous ice and the solid ice, we prescribe

$$A_\mathrm{w} = A_\mathrm{i} \quad \text{at} \quad Z = Z_\mathrm{m}, \tag{46}$$


$$C_\mathrm{w} = C_\mathrm{i} \quad \text{at} \quad Z = Z_\mathrm{m}. \tag{47}$$

With the absorption of shortwave radiation being independent of microbial abundance, the microbe and nutrient problem decouples from the weathering crust problem. The analytical solution to the weathering crust problem found in Sect. 2.3 can first be used to calculate $\phi, m_\mathrm{int}, \dot{h}$ and $Z_\mathrm{m}$. Then, using these results, equations (44) and (45) can be solved (numerically) with
the boundary conditions (46) and (47) for $A_\mathrm{w}$ and $C_\mathrm{w}$.

Before looking at some solutions, we discuss what we expect to be typcial values of the microbial abundance and nutrient concentrations. Motivated by Christner et al. (2018), we choose $A_\mathrm{max} = 10^4$ cells mL$^{-1}$ and $A_\infty = 10^2$ cells mL$^{-1}$ as sensible values for the maximum microbial abundance and the microbial abundance deep in the ice. The nutrient concentration deep in the ice $C_\infty$ gives the maximum value of $C_\mathrm{w}$ in the steadily melting model. To estimate a value for $C_\infty$, we note that Holland
et al. (2019) measured concentrations of dissolved organic and inorganic nitrogen, phosphorous and carbon near the surface of the Greenland ice sheet dark zone varying in orders of magnitude from about 0.1 $\mu$mol L$^{-1}$ to over 100 $\mu$mol L$^{-1}$. The nutrient concentration of interest will be that of the most limiting nutrient, which could potentially be any of these. Therefore, as a ballpark figure, we take $C_\infty = 1$ $\mu$mol L$^{-1}$. In fact, the actual numerical values of nutrient concentration are not particularly important to the model and we will show the nutrient concentration solutions normalised by $C_\infty$.

## 3.3 Results

### 3.3.1 Effect of microbe growth rate

Figure 5 shows an example solution of how the volume-averaged microbial abundance $\phi A_\mathrm{w}$ and nutrient concentration $\phi C_\mathrm{w}$ vary with depth in the weathering crust shown in Fig. 2. The solutions are shown for two different values of microbial growth rate: $\beta_A = 20$ d$^{-1}$ (dark blue line) and $\beta_A = 5$ d$^{-1}$ (grey line). Even in the porous region (above the dotted line showing where
$Z = Z_\mathrm{m}$), there is an inner region where the microbial abundance $A_\mathrm{w}$ and nutrient concentration $C_\mathrm{w}$ are still approximately their deep ice values, and hence $\phi A_\mathrm{w}/A_\mathrm{max} \approx \phi A_\infty/A_\mathrm{max}$ ($= 0.01\phi$) and $\phi C_\mathrm{w}/C_\infty \approx \phi$. Above this, $A_\mathrm{w}$ increases towards the surface and $C_\mathrm{w}$ decreases. This is because there is more shortwave radiation nearer the surface, enabling more microbial growth. In turn, more microbes mean that more nutrient gets consumed. Moreover, we can see that, as we would expect, there are more microbes when the growth rate is higher (dark blue line). The turning point in the nutrient profiles is the point above
which the nutrient concentration $C_\mathrm{w}$ changes significantly from the deep-ice value $C_\infty$, causing the bulk nutrient concentration $\phi C_\mathrm{w}$ to move off the exponential porosity profile.



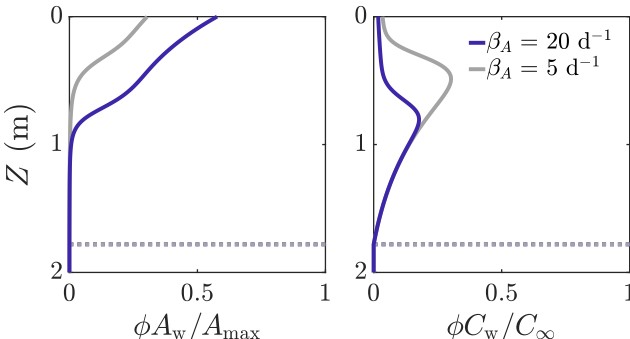

**Figure 5.** Volume-averaged microbial abundance $\phi A_\mathrm{w}$ and nutrient concentration $\phi C_\mathrm{w}$ solutions scaled with the maximum concentrations $A_\mathrm{max}$ and $C_\infty$. Solutions correspond to the weathering crust solutions (dark blue lines) in Fig. 2. The solutions are shown for $\beta_A = 20\ \mathrm{d}^{-1}$ (dark blue line) and $\beta_A = 5\ \mathrm{d}^{-1}$ (grey line). The dotted line shows $Z = Z_\mathrm{m}$. The other parameter values are those shown in Table 1.

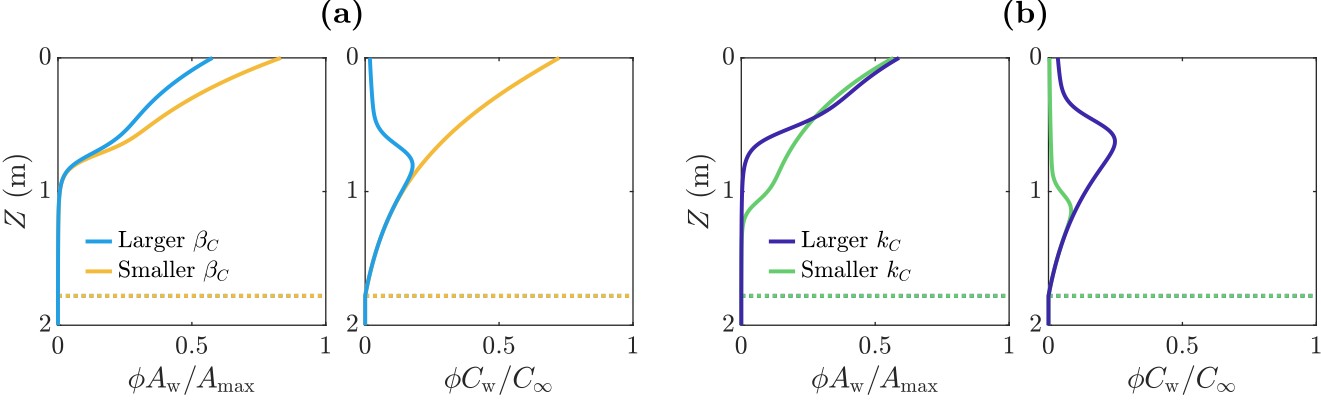

**Figure 6.** Volume-averaged microbial abundance $\phi A_\mathrm{w}$ and nutrient concentration $\phi C_\mathrm{w}$ solutions scaled with the maximum concentrations $A_\mathrm{max}$ and $C_\infty$. Panel (a) shows the solutions for $\beta_C/C_\infty = 10^{-3}\ \mathrm{mL\ cell}^{-1}\ \mathrm{day}^{-1}$ (light blue) and $\beta_C/C_\infty = 10^{-5}\ \mathrm{mL\ cell}^{-1}\ \mathrm{day}^{-1}$ (yellow). Panel (b) shows the solutions for $k_C = 2C_\infty$ (dark blue) and $k_C = 0.2C_\infty$ (green). These solutions correspond to the weathering crust solutions (dark blue lines) in Fig. 2. The dotted lines show the melting depth $Z_\mathrm{m} \approx 1.78$ m. The other parameter values are those shown in Table 1.

### 3.3.2 Effect of nutrient parameters

We now investigate how the distribution of microbes is affected by parameters related to the uptake of the nutrient. In particular, we look at the parameters $\beta_C$ and $k_C$. Firstly, $\beta_C$ is the maximum nutrient uptake rate, with a larger value of $\beta_C$ meaning that 390 the microbes consume more nutrients to achieve the same amount of growth. Figure 6 (a) shows the distribution of microbial abundance $A_\mathrm{w}$ and nutrient concentration $C_\mathrm{w}$ with depth for two values of $\beta_C$ - one 100 times larger (light blue) than the other




(yellow). Consistent with our interpretation of $\beta_C$, the nutrient concentration $C_w$ is lower when $\beta_C$ is larger. Correspondingly, the microbial abundance $A_w$ is lower because the nutrient has been used up more quickly by the microbes, so less nutrient is available for further microbial growth.

The second nutrient parameter $k_C$ is a measure of the point at which the nutrient stops being a limiting factor for the microbes. The larger $k_C$ is, the higher the volume-averaged nutrient concentration $\phi C_w$ must be before the microbial growth rate becomes independent of $\phi C_w$. Equivalently, the smaller $k_C$ is, the lower the volume-averaged nutrient concentration must get before the nutrient becomes a limiting factor. Figure 6 (b) shows the distribution of microbial abundance $A_w$ and nutrient concentration $C_w$ with depth for two values of $k_C$ - one 10 times larger (dark blue) than the other (green). The effect of

decreasing $k_C$ is to increase the size of the region in which there is a significant number of microbes. This is because the porosity decreases with depth, which in turn means the volume-averaged nutrient concentration $\phi C_w$ decreases with depth in the region near $Z = Z_m$ where $C_w \approx C_\infty$. Therefore, as $k_C$ becomes smaller, the depth at which the nutrient becomes a limiting factor (controlled by the size of $\phi C_w$ relative to $k_C$) increases, so the microbes spread deeper into the ice. Note that decreasing $\beta_C$ does not affect the depth to significant microbial growth, because the size of $\beta_C$ does not change the conditions

required for microbial growth - it only affects the rate at which nutrients are consumed.

### 3.3.3  Total microbial abundance

As we did for the weathering crust, we can examine how the steadily melting solution varies with radiative forcing, assuming fixed microbe and nutrient parameters. Firstly, we show solutions for different $Q_0$ and $Q_{si}$ in Fig. 7. Secondly, we look at the dependence of the total amount of microbes in the water $A_{tot,w}$, defined as

$$A_{tot,w} = \int_0^{Z_m} \phi A_w \, dZ. \tag{48}$$

This has units cells m$^{-2}$. As discussed at the end of Sect. 2.3, changing weather conditions (represented by the radiative forcings $Q_{si}$ and $Q_0$ in our model) lead to the formation and removal of the weathering crust. Therefore, we choose to investigate the response of microbes to changes in both $Q_{si}$ and $Q_0$, including positive and negative $Q_0$, to capture the range of conditions experienced by microbes in the weathering crust. For a weathering crust to form, the shortwave radiation $Q_{si}$ must be large

enough. Positive $Q_0$ corresponds to conditions of high turbulent heat flux (warm, cloudy, windy), which promote removal of the weathering crust. Negative $Q_0$ means that the surface would freeze in the absence of shortwave radiation. The observed behaviour is quite different in these two cases.

Figure 8(a) shows that when $Q_0 > 0$ and for small values of $Q_{si}$ - below the critical value required to produce internal melting – there are no water microbes, because there is no water. Then, as $Q_{si}$ increases, $A_{tot,w}$ increases due to the increase in

internal radiation leading to an increase in microbial growth and an increase in the size of the size of the weathering crust $Z_m$ (ie. the region in which microbes can grow). As shown in Fig. 8(c), if $Q_{si}$ is increased further (to less physical values), the total amount of microbes $A_{tot,w}$ starts to decrease again. This is because the internal radiation reaches the level above which it is no longer a limiting factor for the microbes. Increasing the shortwave radiation further does not increase the microbial growth but




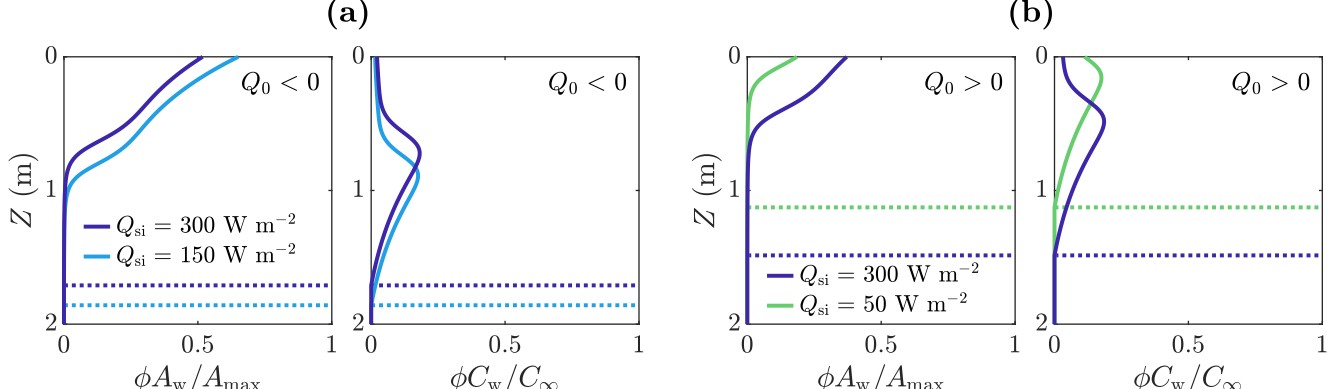

**Figure 7.** The effect of $Q_\mathrm{si}$ and $Q_0$ on volume-averaged microbial abundance $\phi A_\mathrm{w}$ and nutrient concentration $\phi C_\mathrm{w}$ solutions scaled with the maximum concentrations $A_\mathrm{max}$ and $C_\infty$. Panel (a) shows the solutions for $Q_0 = -20$ W m$^{-2}$ (negative) in the case $Q_\mathrm{si} = 150$ W m$^{-2}$ (light blue) and $Q_\mathrm{si} = 300$ W m$^{-2}$ (dark blue). Panel (b) shows the solutions for $Q_0 = 20$ W m$^{-2}$ (positive) in the case $Q_\mathrm{si} = 50$ W m$^{-2}$ (green) and $Q_\mathrm{si} = 300$ W m$^{-2}$ (dark blue). The dotted lines show the weathering crust thickness $Z_\mathrm{m}$. The other parameter values are those shown in Table 1.

does continue to increase the surface lowering and runoff, meaning that the removal of microbes in runoff begins to dominate
microbial growth, causing a reduction in $A_\mathrm{tot,w}$.

However, when $Q_0 < 0$, the total number of microbes $A_\mathrm{tot,w}$ decreases monotonically with increasing $Q_\mathrm{si}$ (compared to increasing when $Q_0 > 0$). The reason for this relates back to Fig. 3. As previously discussed, when $Q_0 < 0$, the melting depth $Z_\mathrm{m}$ decreases as $Q_\mathrm{si}$ increases because surface lowering increases at a faster rate than internal melting. Therefore, the region in which microbes can grow gets smaller, with the microbe distribution shifting upwards, as shown in Fig. 7(a).

Figure 8(b) also shows how $A_\mathrm{tot,w}$ depends on the other surface radiation $Q_0$, which is a combination of net longwave radiation and turbulent heat fluxes, for a fixed value of shortwave radiation $Q_\mathrm{si}$. We see that $A_\mathrm{tot,w}$ is a decreasing function of $Q_0$. This is because increasing $Q_0$ increases the surface melting without affecting the internal melting or microbial growth. Hence the surface runoff increases without microbial growth increasing to compensate. When $Q_0$ gets large enough, there is no time for any microbial growth before the microbes get removed in runoff, so the microbial abundance remains close to the
small background value $A_\infty$.

The combined effect of $Q_\mathrm{si}$ and $Q_0$ on $A_\mathrm{tot,w}$ is shown in the contour plot in Fig. 8(f). In addition to the behaviour captured by Fig. 8(a)-(c), this shows how increasing $Q_0$ to more positive values causes the maximum of the blue $A_\mathrm{tot,w}$ against $Q_\mathrm{si}$ curve in Fig. 8(c) to decrease and move to a higher value of $Q_\mathrm{si}$. As discussed, the total number of microbes $A_\mathrm{tot,w}$ is determined by a balance between the microbial growth in the interior and removal of microbes at the surface in runoff, with the position
of the maximum being controlled by the point past which increasing $Q_\mathrm{si}$ further no longer increases the microbial growth significantly. When $Q_0$ is more positive, this point is delayed to larger $Q_\mathrm{si}$ because the size of the region in which microbial



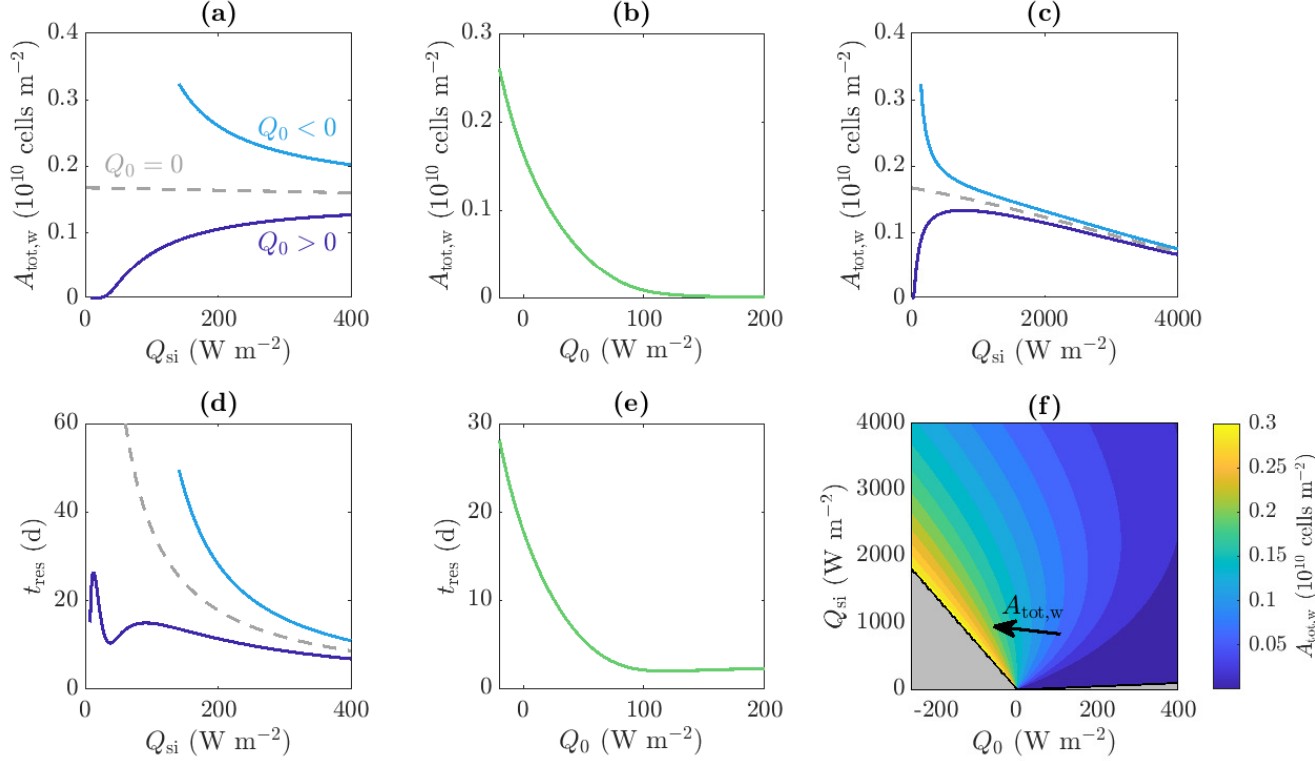

**Figure 8.** Dependence of (a), (b), (c), (f) total algae in the water $A_{\mathrm{tot,w}}$, and (d), (e) approximate residence time of the microbes $t_{\mathrm{res}}$, on the incoming shortwave radiation $Q_{\mathrm{si}}$ and other surface radiation $Q_0$. (a), (c) and (d) show the solutions for $Q_0 = 20$ W m$^{-2}$ (dark blue line), $Q_0 = -20$ W m$^{-2}$ (light blue line) and $Q_0 = 0$ W m$^{-2}$ (grey dashed line). (b) and (e) show solutions with $Q_{\mathrm{si}} = 200$ W m$^{-2}$. (c) is an extension of (a) to larger $Q_{\mathrm{si}}$ values. (f) shows contours of $A_{\mathrm{tot,w}}$ in the region for which steadily melting states with both internal and surface melting are possible. The contour spacing is $0.02 \times 10^{10}$ cells m$^{-2}$. The regions for which there is no surface melting and no internal melting are also shown (grey, see Fig. 3). The other parameter values are those shown in Table 1.

growth is possible, $Z_{\mathrm{m}}$, takes longer to tend to its maximum value (see Fig. 3). Hence, for more positive $Q_0$, $Q$si must get larger before $A_{\mathrm{tot,w}}$ starts decreasing with $Q_{\mathrm{si}}$.

Importantly, the key result from Fig. 8(f) is that the total number of microbes is largest when $Q_0$ is negative and $Q_{\mathrm{si}}$ is just

large enough to allow surface melting, which are the conditions under which weathering crust growth is most favoured.

### 3.3.4 Residence times

Another quantity of interest is the residence time of the microbes - that is, how long the microbes remain in the weathering crust before being removed in runoff. This is of interest because it affects the carbon fluxes from the ice sheet - in particular whether or not the ice sheet is a net source or sink of CO$_2$. Cook et al. (2012) showed the Greenland ice sheet to be in a

stable state of autotrophy (net removal of CO$_2$ from the atmosphere) due to the presence of cryoconite and surface algae, with





increased warming leading to increased biomass and carbon fixation. However, they warned that too much warming could lead to the removal of microbes in runoff, reducing the removal of $CO_2$ from the atmosphere.

In our steadily melting model, we approximate the average residence time by

$$t_{\mathrm{res}} = \frac{A_{\mathrm{tot,w}}}{q},$$ (49)

where

$$q = -\dot{h}\phi A_{\mathrm{w}}|_{Z=0}$$ (50)

is the microbial runoff rate at the surface. The expression (49) for $t_{\mathrm{res}}$ is the ratio between the amount of microbes in the water of the weathering crust and the rate at which the microbes are removed from the weathering crust. Therefore, we can think of $t_{\mathrm{res}}$ as being the average amount of time it would take to remove all the microbes from the weathering crust water. We can also

interpret this as the average amount of time each microbe spends in the weathering crust, ie. the residence time. Figure 8(d) shows the dependence of $t_{\mathrm{res}}$ on the incoming shortwave radiation $Q_{\mathrm{si}}$. This is shown for positive (dark blue line) and negative (light blue line) values of $Q_0$. As with the total number of microbes $A_{\mathrm{tot,w}}$, the observed behaviour is quite different depending on the sign of $Q_0$.

When $Q_0 > 0$, the global behaviour (ignoring the local minimum in Fig. 8(d) which we will discuss shortly) is that as

$Q_{\mathrm{si}}$ increases, $t_{\mathrm{res}}$ initially increases then decreases again, for the same reasons as those just discussed around $A_{\mathrm{tot,w}}$: as $Q_{\mathrm{si}}$ initially increases, the microbial growth rate increases so there are more microbes in the water and these have to wait longer until they can be removed. Once $Q_{\mathrm{si}}$ is large enough to no longer be a limiting factor, the removal of microbes at the surface starts to dominate. This agrees with the previously discussed behaviour expected by Cook et al. (2012): increased warming leads to increased biomass (and residence times), until the microbes get removed by runoff. The local minimum observed in

Fig. 8(d) occurs because the microbial abundance in the ice is small, $A_{\mathrm{i}}/A_{\mathrm{max}} = A_\infty/A_{\mathrm{max}} = 0.01$. As $Q_{\mathrm{si}}$ increases, the microbial abundance $A_{\mathrm{w}}$ initially does not increase very much from $A_\infty$ (seen from the flat section in Fig. 8(a) for small $Q_{\mathrm{si}}$). When significant microbial growth does begin, growth first occurs at the surface, with the surface microbial abundance becoming quite large before the region of significant growth spreads downwards. When near-surface growth dominates, the microbial runoff rate $q$ (which depends on the surface value of the microbial abundance) grows much more rapidly with $Q_{\mathrm{si}}$

than the total number of microbes $A_{\mathrm{tot,w}}$ because the microbial abundance is only growing significantly near the surface. Hence the residence time $t_{\mathrm{res}}$ decreases. Once significant microbial growth begins spreading into the interior, $A_{\mathrm{tot,w}}$ grows more quickly than $q$, so $t_{\mathrm{res}}$ increases again.

However, when $Q_0 < 0$, $t_{\mathrm{res}}$ decreases monotonically with increasing $Q_{\mathrm{si}}$ (as does $A_{\mathrm{tot,w}}$). The residence time decreases for the same reason that it decreases for large $Q_{\mathrm{si}}$ when $Q_0 > 0$ - increasing $Q_{\mathrm{si}}$ increases the surface lowering but does not

increase the growth rate because radiation is high enough that it is no longer limiting. The non-monotonic behaviour for smaller $Q_{\mathrm{si}}$ is missed because $Q_{\mathrm{si}}$ cannot take such small values when $Q_0 < 0$, otherwise surface melting, and hence steadily melting states, would not be possible (see Fig. 3 and surrounding discussion).

Figure 8(e) also shows that the approximate residence time $t_{\mathrm{res}}$ is a decreasing function of $Q_0$ for a fixed value of $Q_{\mathrm{si}}$. This is for the same reasons (already discussed) that $A_{\mathrm{tot,w}}$ is a decreasing function of $Q_0$ for a fixed value of $Q_{\mathrm{si}}$.





### 3.3.5 Summary


Our steadily melting state model has shown that microbial abundance in the weathering crust generally decreases with depth, as expected by Cook et al. (2016) and observed by Christner et al. (2018). This is because the radiation - which the microbes need to grow - decreases with depth. Moreover, we have seen that significant microbial growth often only occurs in a top portion of the weathering crust, with the size of this region being controlled by microbial growth and nutrient parameters.

Furthermore, we have seen that the radiative forcings $Q_{\mathrm{si}}$ and $Q_0$ have a significant impact on the total number of microbes in the weathering crust and their residence time. This impact is via the balance between internal radiation (which favours growth of microbes and the weathering crust) and surface radiation (which increases surface melting and removal of microbes via runoff). In general, increasing internal radiation at a greater rate than surface radiation leads to an increase in the number of microbes and their residence time. In line with this, we observe different behaviour depending on the size of the shortwave

radiation $Q_{\mathrm{si}}$ and the sign of the combined longwave radiation and turbulent heat fluxes $Q_0$, suggesting that the behaviour of the microbial population varies during the changeable conditions that lead to the formation and removal of the weathering crust. (Recall that $Q_{\mathrm{si}}$ must be large enough for weathering crust formation to occur and $Q_0 > 0$ favours removal (Muller and Keeler, 1969).)

## 4 Feedbacks between microbe growth, albedo, and melting

The weathering crust and the microbes within it give rise to some potential albedo feedbacks that can lead to amplified reductions in albedo and increases in melting. Here we will discuss the melt-albedo feedback (e.g. Box et al., 2012) and the microbe-albedo feedback (e.g. Tedstone et al., 2020; McCutcheon et al., 2021) and explore the ability of our model to capture and quantify these feedbacks. We first briefly outline the potential feedbacks.

As the surface of an ice sheet melts, the surface transitions from snow covered to bare ice to having meltwater on the surface.
The change in surface type from snow-covered to water-covered leads to a reduction in the albedo. This in turn means that more radiation is absorbed so more melting occurs, reducing the albedo further. The result is a positive feedback: the melt-albedo feedback (e.g. Box et al., 2012).

The surface of ice sheets can also be made darker by biological factors, especially glacier ice algae and cryoconite (Hotaling et al., 2021), with glacier ice algae thought to be a key contributor to the darkening of the Greenland ice sheet (Benning
et al., 2014). There is thought to be a potential positive feedback between ice algae and meltwater, whereby the presence of algae on the ice sheet surface lowers the albedo, increasing melting, which releases nutrients that had been frozen in the ice, enabling more algal growth (e.g. Tedstone et al., 2020; McCutcheon et al., 2021). The darkening effect is particularly strong for surface ice algae because the algae have a special dark pigment to protect them from high levels of UV and photosynthetically active radiation (Williamson et al., 2019, 2020). However, a similar, but weaker, feedback can be expected for microbes in the
weathering crust. Furthermore, the framework we provide here for studying microbe-albedo feedback in the weathering crust can be easily adapted to ice algae.





### 4.1 Model

To introduce the melt-albedo and microbe-albedo feedbacks into our steadily-melting model, we make the absorption coefficient $\alpha$ a linearly increasing function of either the porosity $\phi$ or the bulk microbial abundance $A_{\mathrm{bulk}} = \phi A_{\mathrm{w}} + (1 - \phi) A_{\mathrm{i}}$. This captures the fact that more shortwave radiation is absorbed when there is more water (compared to ice), or when there are more microbes. The albedo $a$ is then calculated as an output of the model from the additional boundary condition (22), as discussed in Sect. 2. The absorption coefficient functions we chose are

$$\alpha = \alpha_0 \Big(1 + 3\phi\Big) \tag{51}$$

to capture the melt-albedo feedback, and

$$\alpha = \alpha_0 \Big(1 + 3(A_{\mathrm{bulk}} - A_\infty)\Big) = \alpha_0 \Big(1 + 3\phi(A_{\mathrm{w}} - A_\infty)\Big) \tag{52}$$

to capture the microbe-albedo feedback, where we have used that $A_{\mathrm{i}} = A_\infty$ in our steadily-melting setup. Here, $\alpha_0$ is the absorption coefficient of pure ice containing the small background abundance of microbes $A_\infty$. The specific forms of (51) and (52) have been chosen as a simple way to include the feedbacks in our model.

### 4.2 Results

By making the absorption coefficient a function of the porosity $\phi$ and/or the microbial abundance $A_{\mathrm{w}}$, the steadily melting weathering crust problem can no longer be solved analytically. Furthermore, a dependence on $A_{\mathrm{w}}$ fully couples the microbe and nutrient problem to the weathering crust problem - we can no longer solve the weathering crust problem on its own first. Therefore, we must solve the full problem numerically.

As expected, making the absorption of shortwave radiation increase with water content or microbial abundance causes a reduction in albedo in our model (Fig. 10(c)). We find that including the meltwater coupling (51) reduces the albedo to 0.47 compared to 0.6 in the uncoupled case with $\alpha = \alpha_0$. Similarly, the microbe coupling (52) reduces the albedo of the steadily melting solution to 0.51 from 0.6.

Figure 9 shows a comparison between the porosity, internal radiation and microbial abundance distributions in the coupled and uncoupled cases. The effects of including meltwater coupling and microbe coupling are similar. In both cases, more radiation is absorbed by the ice than in the uncoupled case (seen in the reduction in albedo). However, the internal radiation does not increase everywhere - it is higher than the uncoupled case near the surface but lower at depth. This is because there is more water and more microbes near the surface, so absorption of radiation is greatest near the surface, which in turn means that there is less radiation left to penetrate deeper into the ice.

The overall effect of including the coupling is to decrease the size of the melting region (decrease $Z_{\mathrm{m}}$), lower the porosity and reduce the total amount of microbes (Fig. 9). This might seem counterintuiutive; the feedbacks previously discussed would suggest that making the absorption depend on water content or microbial abundance would lead to an increase in water content and microbial abundance, since more water and microbes leads to greater absorption resulting in even more melting





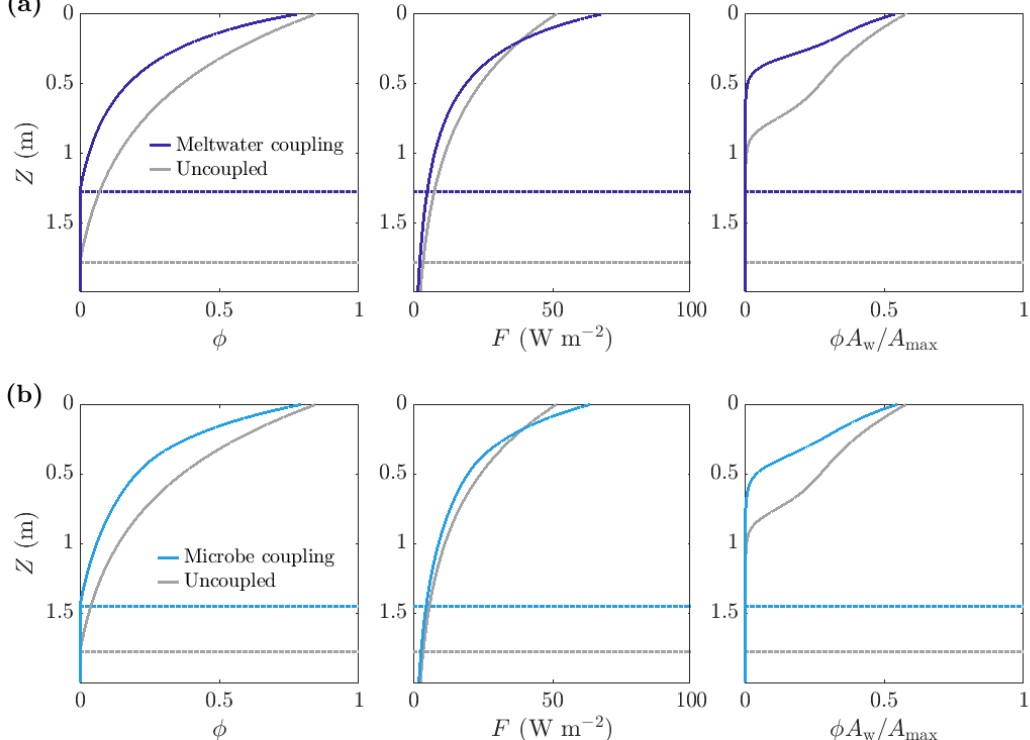

**Figure 9.** Comparison of steadily melting solutions with albedo feedbacks (dark and light blue) and without (grey). (a) shows meltwater coupling, where $\alpha = \alpha_0(1 + 3\phi)$, so that the absorption increases with water content. The albedo in the coupled case is 0.47, compared to 0.6 in the uncoupled case. (b) shows microbe coupling, where $\alpha = \alpha_0(1 + 3\phi(A_{\mathrm{w}} - A_{\infty}))$, so that the absorption increases with the bulk microbial abundance. The albedo in the coupled case is 0.51. The dotted lines show the melting depth $Z_{\mathrm{m}}$. The parameter values are those shown in Table 1.

and microbial growth. However, we are observing the previously discussed removal of microbes in runoff, proposed by Cook et al. (2012). The increase in absorption has led to an increase in surface melting, lowering and runoff, with enhanced removal
of microbes in runoff dominating any increase in microbial growth. Therefore, including the coupling leads to the shifting upwards of the microbial abundance distribution we saw in Sect. 3.3.2.

Figure 10 reproduces the plots of total number of microbes in the water $A_{\mathrm{tot,w}}$ and residence time $t_{\mathrm{res}}$ as a function for $Q_{\mathrm{si}}$ from Fig. 8, this time showing the results for meltwater coupling and microbe coupling as well as the uncoupled case from Fig. 8 (grey). The variation with $Q_{\mathrm{si}}$ for $Q_0 < 0$ (top row) and $Q_0 > 0$ (middle row) is shown, as well as the variation with $Q_0$
(bottom row). These show that including the meltwater or microbe coupling reduces the total number of microbes and reduces their residence time, with the main effect when $Q_0 > 0$ being to reduce the second local maximum. This is because the region of significant algal growth is smaller with coupling (Fig. 9), so the increase in $t_{\mathrm{res}}$ from the local minimum to the second local maximum, which corresponds to the spread of the significant microbial growth region deeper into the ice (discussed around Fig. 8), is reduced.





**Figure 10.** Dependence of (a), (d), (g) total algae in the water $A_{\text{tot,w}}$, (b), (e), (h) approximate residence time of the microbes $t_{\text{res}}$, and (c), (f), (i) albedo $a$ on the incoming shortwave radiation $Q_{\text{si}}$ ((a)-(f)) and other surface radiation $Q_0$ ((e)-(g)), in the uncoupled (grey), meltwater coupling (dark blue) and microbe coupling (light blue) cases. The value of $Q_0$ is $-20$ W m$^{-2}$ (negative) in (a)-(c), and $20$ W m$^{-2}$ (positive) in (d)-(f). The value of $Q_{\text{si}}$ is $200$ W m$^{-2}$ in (g)-(i). The other parameter values are those shown in Table 1.

Note that Fig. 10 shows that the total number of microbes, residence time and albedo are all lower with meltwater coupling than microbe coupling, but this should not be read into too much. The forms of the absorption coefficient functions (51) and (52) are illustrative only and could easily be modified such that the the microbe coupling has a larger effect (e.g. by replacing 3 in (52) with a larger value). The main point to take away from this investigation is that including coupling between meltwater,





microbes and radiation absorption can lead to a reduction in albedo, but also a reduction in the size of the weathering crust, 565 and its total microbial abundance.

## 5 Discussion and conclusion

We have presented a mathematical model for the weathering crust and the microbes within it, as well as a nutrient for the microbes. We have focussed on one-dimensional steadily melting solutions, where the internal and surface melt rates balance to produce a weathering crust of a steady size. We have assumed that all surface meltwater runs off, removing any microbes or 570 nutrients that were in the water. In reality, the lateral flow of meltwater across the ice sheet surface and through the weathering crust is important for the transport of nutrients and microbes across the ice (e.g. Stibal et al., 2012), but our one-dimensional model is unable to capture this.

We have made a number of simplifications in order to produce a reduced problem from which we can gain informative insights. One simplification of reality is our treatment of the microbes and their nutrients. We have assumed that there is a single 575 type of phototrophic microbe and a single representative nutrient for the microbe. In reality, the weathering crust contains a range of different microbes: primary producers (e.g. algae and filamentous cyanobacteria) which use solar radiation as their energy source, and heterotrophs which use autochthonous organic carbon produced by the primary producers as their energy source (Stibal et al., 2012). Our model could be thought of as describing a single representative type of primary producer, which requires both solar radiation and nutrients for growth. Alternatively, it could be interpreted as modelling (crudely) the combined 580 influence of all microbes. Clearly there are more complex interactions that could be added, at the expense of introducing further parameters and potentially losing interpretability.

Similarly, the reality is that microbes require multiple nutrients to survive and grow, namely carbon (C), nitrogen (N) and phosphorous (P) (Hessen et al., 2013). Our representative nutrient can be thought of as whichever of these three nutrients is most limiting. In the case of surface glacier ice algae, McCutcheon et al. (2021) have shown that phosphorous is the limiting 585 nutrient.

Our model has given insight into the porosity and microbe distributions in the weathering crust. We have found the microbial abundance in the weathering crust generally decreases with depth. This agrees with the expectation of Cook et al. (2016) and the observations of Christner et al. (2018). Since radiation decreases with depth, it is expected that there will be more primary producers (with protective pigments) near the surface and more heterotrophs (which are less light-adapted) deeper down (Cook 590 et al., 2016). The decrease in radiation with depth also leads to a decrease in porosity and melting with depth.

Furthermore, we have observed that the climate forcing (represented by the shortwave radiation $Q_{\mathrm{si}}$ and combined longwave radiation and turbulent heat fluxes $Q_0$ in our model) has a clear impact on the size of the weathering crust and the residence time and abundance of microbes in the weathering crust. Our model shows that it is the balance between internally absorbed radiation and surface forcings that controls these quantities. The relationship is complex, but, in general, increasing the surface 595 forcings relative to internal radiation (for example, increasing the turbulent heat fluxes and longwave radiation $Q_0$) causes the size of the weathering crust, the residence time and the total number of microbes in the crust to all decrease. This is due to



enhanced surface melting and runoff causing removal of the weathering crust and the microbes within it. This is in line with the observation that warm, windy, overcast conditions (high turbulent heat fluxes) favour weathering crust removal (Muller and Keeler, 1969), as well as the suggestion that excessive warming could lead to a removal of microbes in runoff, threatening the

Greenland ice sheet's state of autotrophy (net removal of $CO_2$) (Cook et al., 2012).

Our model also allows for the investigation of positive feedbacks. We have explored melt-albedo and microbe-albedo feedbacks by making the absorption of shortwave radiation a function of the water and microbe content of the weathering crust, respectively. These feedbacks lead to enhanced melting of the ice sheet surface. Our model suggests that both feedbacks act to reduce the weathering crust size and number of microbes in the weathering crust, as well as their residence times. This is due to

enhanced melting leading to an increased runoff rate, so more microbes get removed from the surface. Hence these feedbacks are self limiting - the reduction in albedo cannot keep going unlimited because the increased melting will leads to the removal of the microbes and meltwater that cause the albedo reduction.

In conclusion, we have shown that a mathematical model can be used to capture and explore key characteristics and behaviour of the weathering crust and the microbes within it. In particular, even with a number of simplifications, it shows that the

response of the weathering crust and its microbial community to changes in atmospheric forcing is complex, and can lead to both increases and decreases in microbial abundance. We acknowledge that the model has crudely approximated many aspects of the dynamics, but hope that this study can provide a framework into which more detailed biogeochemistry and physics can be incorporated in the future.

*Code availability.* The MATLAB code used to produce the results and figures in this article is made available at https://doi.org/10.5281/zenodo.7199159.

*Author contributions.* TW and IH developed the model. TW wrote the code and produced the results. TW wrote the paper with input from IH.

*Competing interests.* The authors declare that they have no conflict of interest.

*Acknowledgements.* TW acknowledges the support of an EPSRC doctoral studentship. TW and IH would like to thank Liz Bagshaw and Chris Williamson for useful discussions about microbes.



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
