# Peer review of "A model of the weathering crust and microbial activity on an ice-sheet surface"

_EGUsphere, 2022_

## Author Response (AR1)

**Response to Andrew Tedstone reviewer comments**

Thank you very much for your kind words and constructive comments which have allowed us to improve our manuscript. We really appreciate the time you have put into writing the review, and we are glad that you enjoyed reading it.

Below is a copy of your comments with our responses written below each one.

**Reviewer comment:** This article presents what is to my knowledge the first model to capture two/three bare-ice processes simultaneously: (i) the development and destruction of the porous near-surface weathering crust; (ii) the growth of microbes (e.g. ice algae) within that crust; and (iii) nutrient availability within the crust. In doing so it synthesises and 'formalises' several field-based studies from the last ~50 years. The model's behaviour corresponds well, at least at first order, to field observations, and it yields further process-based insight. These insights are especially strong in revealing the inter-play between (a) shortwave radiation versus other radiation sources for crust development and microbe growth; and (b) the impacts of radiation upon melt and in turn runoff of microbes and nutrients.

I commend the authors on this timely and thorough study of a complex topic. It's written concisely and with excellent figures. I enjoyed reading it. I caveat that as my knowledge lies rather more in field observations of these processes than mathematics, I cannot formally assess the suitability of the numerical methods employed, and so I have restricted my comments accordingly. Within this context, overall I find this study to be in excellent shape and I have only a few minor comments, mainly concerning the discussion/wider applicability of the model.

**Response:** Thank you for your positive words. Coming from a mathematical background, we appreciate your insight into field observations.

Minor comments:

**Reviewer comment:** I would really like to see this model run for a full melt season at a location such at the SW GrIS 'dark zone' where these processes are known to be important. However, I appreciate that this is almost certainly too much work and content for the present study, so instead it would be very useful to at least comment on the feasibility of such a temporal model suite in the Discussion/Conclusions. I make this comment partly in the context that bare ice albedo schemes in the major regional climate model surface schemes are very simple, often yielding quite poor comparison with in-situ observations (e.g. Fettweis et al., 2017, The Cryosphere) and so the search is on for more physical approaches that yield closer correspondence with observed albedo.

**Response:** We agree that it would be interesting to study evolution over a melt season, and also agree that this would be too much work for the current study. However, this is certainly

something we hope to investigate in future. Moreover, thank you for the literature suggestion. We have included a discussion of temporal modelling and mentioned the relevance to bare-ice albedo schemes in the Discussion and Conclusion.

**Reviewer comment:** Shortwave radiation attenuation with depth: Cooper et al. (2020, The Cryosphere) present the first observations to my knowledge of light attenuation through a weathering crust. I did not see this study referenced in the present m/s. Please consider commenting on how their observations compare to the choices from Hoffman et al. (2014), Taylor and Feltham (2005).

**Response:** We had not come across the Cooper et al paper before - thank you for the suggestion. It was an interesting read. We have now included a comment on their observations in Sec. 2.3 and justified why we have chosen to follow the Taylor and Feltham (2005) choice of extinction coefficient.

**Reviewer comment:** Microbial abundance and its interaction with runoff: Overall, I concur with the approach taken here. I agree that the parameter A_max is basically a reasonable choice. However, with surface ice algal abundances in excess of 10,000 cells ml-1 reported previously for south-west Greenland (e.g. Cook et al., 2020, The Cryosphere, Wang et al., 2018, Geophysical Research Letters), I think some consideration of how the surface can support such high abundances is still warranted. Specifically, I wonder if the instant microbial runoff here is realistic.

To my understanding, on the basis of the modelling in the present study, then we would expect the high growth rates in large melt years to be offset by widespread microbial runoff - yet we see that in large melt years then we get high persistent algal abundance, implying that the cells can persist at the surface. I'm not sure whether the mechanisms by which algal cells can persist at the surface have been identified by the microbiology community, so there is probably a knowledge gap here. Nonetheless, I am of the view that currently the study provides rather an estimate of the microbial abundance within the weathering crust, but not the total 'system' abundance including algae also 'stored' on the surface of the weathering crust as could be implied at lines 564-565.

**Response:** Our understanding is that there are some microbes that live on the surface (eg. in the top 2cm) of the weathering crust (ice algae) and others that live in the weathering crust aquifer below the surface, with the surface microbes (ice algae) being able to sit on the ice surface, and the aquifer microbes getting carried around in the meltwater. For simplicity, we chose to focus on the microbes being transported in the weathering crust aquifer, neglecting the surface ice algae for now. This is partly why we chose instant microbial runoff, since we are assuming the microbes in our study move with the meltwater, unable to attach themselves to the ice. Your comment suggests we ought to make this clearer in our manuscript - thank you. We have added clarification on these points at the start and end of Sec. 3.1.

We also acknowledge that neglecting ice algae makes our current model less useful for making significant claims about the albedo, since ice algae provide a major contribution to this. However, we wanted to demonstrate the kind of behaviour that can result from a coupling between microbes, melting and albedo, so that our study can be used as a starting point for further investigations that include surface ice algae too. To improve our study, we could write a separate model (which could be coupled to the current model) for the evolution of surface ice algae which can persist in the presence of runoff, but this is an area for future work. We have added a discussion of this to Sec. 5.

**Reviewer comment:** Similar to my comment about behaviour through a full melt season, I would welcome some brief discussion about how the model could capture (or not) the spatio-temporal dynamics of algal blooms and weathering crusts.

**Response:** The model definitely has the potential to be extended to consider spatio-temporal dynamics. We have added a discussion of this to Sec. 5.

**Reviewer comment:** Literature suggestion: the authors might not be aware of Schuster's (2001) PhD thesis, 'Weathering crust processes on melting glacier ice (Alberta, Canada)'. This could be worth considering, in particular because it contains the only other significant attempt to model the weathering crust that I'm aware of.

**Response:** Thank you for the suggestion. This thesis proved very relevant, providing both qualitative and quantitative observations of the weathering crust, as well as a model of the vertical density profile. We have now compared our work to Schuster's results at various points throughout the manuscript, including Sec. 2.1, Sec. 2.2, and Sec. 5.

**Response to Sammie Buzzard reviewer comments**

Thank you very much for taking the time to review our manuscript. Your comments have been very helpful in drawing our attention to areas to be clarified and improved.

Below are our responses to each of the comments in turn.

**Reviewer comment:** Firstly, I'd like to commend the authors for making the first steps in tackling this difficult modelling problem- this paper will definitely be of use to the community and an important building block for future modelling work. It is also an excellent example of the power of 1-D models and demonstrates multiple examples of things we can learn from these and I enjoyed reading it. The paper is mostly clearly written, with the equations explained in an accessible manor. On this basis I would recommend this paper for publication, but with some suggested changes as detailed below, with one main criticism that would need addressing first.

**Response:** Thank you for your kind words and constructive criticism. We are glad you enjoyed reading the manuscript.

**Reviewer comment:** The main addition I would suggest for this paper is the need for better validation of the results. While I absolutely appreciate that this is a simplified model and not intended to replicate reality exactly (and the authors are clear about this), it is still useful and important to understand how close to reality the results are. This is a topic with limited field observations, but even some rough numbers for context to check values are within the correct order of magnitude would much improve the paper. For example, there is one mention of the depth of the weathering crust being 'up to 1 m thick' in the introduction but no discussion of the depth of your modelled weathering crusts, or if this 1 m is a likely value, or an upper end of the scale. This may also be useful in addressing some of the choices of values you have made, some of which are mentioned below.

**Response:** We agree that we had not done enough of this and have added more comparison and discussion of the results. We have changed the phrase 'up to 1m thick' to 'around a metre thick' to make it clearer and more consistent with the following observations. Cooper et al 2018 (Cryosphere) offer some useful, but limited, observations of the weathering crust depth in Greenland using ice cores. They observe that the weathering crust in their study region is at least 1 m thick, and could be around 2 m thick. Our results broadly agree with this observation. We also reproduce the non-linear decrease of porosity with depth observed by Cooper et al 2018 and Schuster 2001 (thesis). However, our surface porosity values are larger than those observed by Cooper et al 2018 and Schuster 2001 (~0.85 compared to ~0.65). This could be due to the fact that our model assumes that the entire crust is saturated, neglecting the observed unsaturated surface layer. The presence of near-surface water in our model leads to increased absorption and melting compared to if the near-surface was unsaturated. Furthermore, we would expect the crust to disintegrate once the porosity gets above a certain limit, but this is not included in our model. We have added comparisons to observations from Cooper et al (2018) and Schuster (2001), and discussions of the above ideas to Sec. 2.3 and Sec. 5.

**Reviewer comment:** Here are some minor line by line comments:

**Reviewer comment:** Line 16: The initial description of what a weathering crust is a little brief and could be more informative. As this is the first attempt to model something like this the reader could be helped to understand this structure a little more e.g. with a figure, or a more detailed description as it's not guaranteed they will know about weathering crusts in detail. The EGU Cryosphere blog on this topic last year was a good explainer https://blogs.egu.eu/divisions/cr/2021/08/20/did-you-know-about-the-weathering-crust-five-things-you-never-knew-about-glacier-surfaces/

**Response:** Thank you for pointing this out. We have added a new first figure with a photo and schematic of the weathering crust, as well as slightly expanding our initial description in the introduction.

**Reviewer comment:** Line 43: You don't really discuss if this assumption is realistic e.g. do we see these kind of conditions over the timescales that you run your model for?

**Response:** The steadily melting states we have chosen to focus on correspond to an equilibrium between internal melting and surface lowering. Schuster 2001 observed that this equilibrium can occur on cloudy days when the crust is transitioning between growth and decay. However, it is not expected that this equilibrium lasts long. The weathering crust is very changeable, growing and decaying over a few hours as weather conditions change between clear skies (favouring growth) and warm, windy, cloudy conditions (favouring crust removal). Therefore, it would be preferable to have a time-dependent model which can capture this variation. However, for simplicity and to gain some initial insight, we chose to use steady forcings. We have included this discussion where we introduce the steadily melting solutions in Sec. 2.2.

**Reviewer comment:** Line 68: What values did you use for u_i, u_w, etc? Why not use Darcy's Law?

**Response:** We end up assuming that u_i and u_w are both zero in our steadily melting model. If we were to include water flow (non-zero u_w), we would use Darcy's law. The ice velocity u_i could be non-zero if accounting for the upward advection of ice in the ablation zone, in which case we would prescribe u_i. We have slightly reworded the relevant section to clarify this point.

**Reviewer comment:** Line 103: The choice to make these velocities upwards didn't make sense to me here as water is draining downwards. Maybe this just isn't worded well but it seemed counterintuitive.

**Response:** We chose to use upward velocities (which, in the case of the water, we expect to be negative) to be consistent with using z as the upward vertical coordinate. We have added clarification that we expect the (upward) water velocity to be negative.

**Reviewer comment:** Figure 1- not a comment, I just wanted to say this was really well produced in that it was clear and helped my understanding of the model setup a lot.

**Response:** Thank you for the positive comment - we're glad you found the figure informative.

**Reviewer comment:** Lines 159-161: This sentence doesn't quite make sense. I can see what you're getting at from the figure but this needs rewording.

**Response:** We have reworded this sentence to make it clearer.

**Reviewer comment:** Line 208: Where did this definition come from?

**Response:** Solving the linear system of equations (12)-(13) gives a solution looking like exp( - (\alpha^2 + 2*\alpha*r)^½ *Z ). When we have exponentially decaying radiation, the extinction coefficient \kappa is defined such that the radiation looks like exp( - \kappa*Z ). Equating the two gives our 'definition', but it would be more accurate to refer to it as a relationship rather than a definition. We have made this change and have also cited Taylor and Feltham 2004, 2005, who previously used this relationship.

**Reviewer comment:** Line 223: This part was unclear. If there is no heat flux at the surface how is melting occurring?

**Response:** We should have said that there is no *conductive* heat flux at the surface (ie. k*dT/dZ=0). Melting can still occur due to the radiative fluxes Q_si and Q_0. We have added the word 'conductive' to the sentence.

**Reviewer comment:** Line 326: This is beyond my expertise, but I wasn't sure if the radiation used for photosynthesis was related to that used for weathering crust formation, or if these are different parts of the spectrum (i.e. are the equations for energy available for growth independent of those available for melting ice?). You say microbe abundance and radiation absorbance are independent on line 362 but don't really explain it, and I imagine most readers aren't going to know a lot about both microbes and radiation.

**Response:** Photosynthesis uses radiation of wavelength 400-700 nm. The radiation absorbed by the ice (used for weathering crust formation) mostly has wavelength >500nm (Cooper et al 2021, Cryosphere). That is, slightly different parts of the spectrum are used for the two different processes, but there is overlap, and both lie within the shortwave (solar) radiation region (wavelengths ~300-3000nm). In our model, we do not consider a continuous spectrum of different wavelengths - we instead use F to represent all wavelengths of shortwave radiation. Therefore, given that the ranges of wavelengths used for photosynthesis and weathering crust formation both lie within the shortwave range, we are justified in using our broadband F in both the equations for microbial growth and for melting ice. The fact that only part of the shortwave spectrum can be used for photosynthesis is included through the constant \alpha_{PAR} in the microbe equation. We have clarified this in Sec. 3.1.

Also, we make the assumption that the radiation absorbance is independent of microbial abundance in Sec. 3.2 for simplicity, but later relax this in Sec. 4 when investigating feedbacks. We have clarified this point when making this assumption in Sec. 3.2.

**Reviewer comment:** Line 379: Why the choice of 5 and 20 here?

**Response:** The choice of $\beta_A = 5$ d^-1 was based roughly on observed doubling times of 3.75 days (Williamson et al 2018) and 5.5 days (Stibal et al 2017) of algal blooms on Greenland. Assuming that all else is in the microbial abundance equation (39) is constant and keeping only the time derivative and growth terms, we find that $\phi A_w \sim \exp(\beta_A*\bar{g}*t)$, where $\bar{g}$ is the depth-average of the growth rate g given in equation (40). We approximate $\bar{g}$ as $g(\bar{F}, \bar{\phi}*\bar{C_w})$ using realistic values for F, $\phi$ and $C_w$, then we can relate the doubling time $t_d$ to $\beta_A$ via $t_d = \log(2)/(\bar{g}*\beta_A)$. (Note that the doubling time of $e^{(c*t)}$ is defined as $t_d = \log(2)/c$ .) $\beta_A = 20$ d^-1 was simply chosen as a value that gives interestingly different behaviour.

It is worth noting that our approximations here were based on observations of doubling times for laterally spreading of algal blooms on the ice surface, not microbial growth in the weathering crust aquifer. Therefore, our approximations are rather crude, but at least serve to demonstrate the key behaviour that it is possible to capture with our model.

We have included both of these points in Sec. 3.3.1.

**Reviewer comment:** Section 4.1: As you later mention this is now a problem that has to be solved numerically- details are missing of how you did this.

**Response:** The solution below the weathering crust (in $Z > Z_m$) was solved analytically. The solution in $Z < Z_m$ was found using the boundary value problem solver bvp4c in MATLAB. We have now mentioned this in the manuscript, in both Sec. 3.2 and Sec. 4.2. The code we used has been made publicly available (see 'code availability' towards the end of the manuscript).

**Reviewer comment:** Equation 51: Why a value of 3 here?

**Response:** We chose 3 fairly arbitrarily. We are not aware of any literature detailing how we would expect the absorption coefficient to depend on the porosity, so we chose to use the linear relationship (51). Instead of 3, we could have chosen any other positive value (we have tried out a few) to give an increase in absorption with increasing water (porosity). The qualitative behaviour, which is what we are interested in here, would remain the same. We now mention this in Sec. 4.1.

**Reviewer comment:** Line 551: Is upwards here in reference to direction or number of microbes? (I suspect the first but it could be clearer).

**Response:** Yes, the 'upwards' is in reference to the direction. We have clarified this.

---

## Author Response (AR3)

**Response to editor comments**

Thank you for recommending our manuscript for publication and for the time you have put into editing it.

We have made the following modifications in response to your comments:

**Editor comment:** L104: 'Similarly for F−.' is stranded. Is this an error, or if deliberate, can it be combined into a sentence?
**Response:** This has been replaced with 'A similarD argument applies for F_-'.

**Editor comment:** L329: Change to 'multiple sources of such nutrients located within the modelled space' - to make clear that the 'nutrient' is rarely a single entity but likely bound to a particle.
**Response:** This change has been made.

**Editor comment:** L580: 'is more water and more microbial biomass near the surface' for grammatical correctness.
**Response:** This change has been made.

**Editor comment:** L623: 'namely C, N, P and a variety of micronutrients. Our model can be interpreted as whichever of the three macronutrients (C, N or P) is limiting. In the case of glacier ice algae and cryoconite microbial communities, P is usually the limiting nutrient (McCutcheon, 2021, Stibal 2008 10.1029/2007JG000429 and Bagshaw 2013 10.1657/1938-4246-45.4.440)'.
**Response:** This change has been made. Thank you for the literature suggestions.

**Editor comment:** Remove at least one or two 'moreovers' from the text - it is rather overused.
**Response:** We have removed three 'moreovers' from the text: L418, L523 and L632 (where we have replaced 'Moreover, we have found the…' with 'We have also found that the…'). Note that the line numbers refer to the marked up manuscript from the previous revision.